# INDEXGUARD: Index-only Backdoor Vetting for Secure Federated PEFT of Large Language Models

Javad Dogani [1]    Devriş İşler [1]    Nikolaos Laoutaris [1]

## Abstract

Federated parameter-efficient fine-tuning (PEFT) enables customizing large language models on private data, yet it is vulnerable to backdoor poisoning—especially when privacy constraints prevent inspection of per-client real-valued updates. We exploit the intuition that poisoning leaves a similar backdoor imprint in which adapter coordinates become salient, so overlap in salient-index supports remains informative even without values. We introduce INDEXGUARD, an unsupervised index-only vetting primitive in which clients send only Top-$K$ salient update indices and the server operates on the induced overlap geometry, clustering clients and filtering cohesion-outlier groups before aggregation. We analyze support stability under bounded rescaling and separability under shared-trigger poisoning under non-IID drift. Across attacks, backbones, and PEFT variants, INDEXGUARD provides end-to-end mitigation, preserving clean accuracy while achieving performance comparable to centralized methods.

## 1. INTRODUCTION

Large language models (LLMs) power many NLP, reasoning, and code-centric applications (Chowdhery et al., 2023; Roziere et al., 2023). Adapting LLMs to domain data, style, and safety needs typically requires fine-tuning (Touvron et al., 2023), yet full model training at modern scales (e.g., 7B–70B+) is increasingly prohibitive in GPU memory (including optimizer states), time, and operational cost (Touvron et al., 2023; Brown et al., 2020). Parameter-Efficient Fine-Tuning (PEFT) mitigates this by freezing the pretrained backbone and training only lightweight adaptation parameters. A common PEFT method is LoRA, which inserts low-rank trainable modules into Transformer

[1]IMDEA Networks Institute, Madrid, Spain. Correspondence to: Javad Dogani <javad.dogani@networks.imdea.org>.

*Proceedings of the 43rd International Conference on Machine Learning*, Seoul, South Korea. PMLR 306, 2026. Copyright 2026 by the author(s).

layers (Hu et al., 2022). LoRA often matches full fine-tuning quality with far fewer trainable parameters and lower optimizer-state memory/overhead (Lester et al., 2021).

Meanwhile, high-quality task and domain data—the main driver of downstream performance—are increasingly sensitive, regulated, and proprietary (Touvron et al., 2023; Kairouz et al., 2021), limiting centralization in real deployments, particularly cross-silo settings such as healthcare (Rieke et al., 2020). Federated learning (FL) therefore emerges as the natural framework for LLM customization under local data control (Li et al., 2020; McMahan et al., 2017). In each round of federated PEFT, clients fine-tune only adapters (e.g., LoRA) on private data with the backbone frozen, and exchange adapter updates rather than raw data or full weights to reduce optimizer-state memory and *often* lowers per-round communication. The server then aggregates them in a FedAvg-style manner (Zhang et al., 2024b; Hu et al., 2022; Zhang et al., 2024a) or adopts adapter-aware, heterogeneity-robust aggregation (e.g., to handle rank mismatch and reduce aggregation noise) (Wang et al., 2024).

Although FL keeps raw data local, it does not eliminate leakage. FL introduces two orthogonal trust axes. **i) Confidentiality**: even without raw-data sharing, federated training can leak whether a record participated (membership inference) or reveal memorized content (extraction) (Shokri et al., 2017; Nasr et al., 2019; Carlini et al., 2021). This is particularly acute in on-device and cross-silo deployments operating on sensitive text, medical records, and behavioral logs, motivating LoRA-tailored DP and cryptographic secure aggregation/homomorphic encryption (HE) to shield shared updates (Liu et al., 2025; Bonawitz et al., 2017; Kairouz et al., 2021). **ii) Integrity**: backdoors can be implanted when a few clients poison adapter updates (or benign clients train on locally poisoned corpora introduced through data collection/annotation pipelines), yielding clean-input utility but trigger-controlled outputs; PEFT amplifies this risk because small, structured adapters enable high-impact, low-footprint manipulation with weak server-side signals.

Despite progress in *centralized* backdoor detection for PEFT, reliable defenses for *federated* PEFT remain largely open. PEFTGuard (Sun et al., 2025) detects PEFT backdoors via supervised binary classification over fea-

tures extracted from *real-valued* self-attention adapter weights, training a meta-classifier on centralized PAD-Bench. AlignIns (Xu et al., 2025) likewise requires unshielded per-client updates and targets full-model FL, so its fit for low-rank, adapter-only training is unclear. These methods inherently assume access to unshielded per-client pre-aggregation adapters—precisely what secure aggregation/privacy-oriented FL hides. Meanwhile, malicious LoRA modules are easily shared and merged in model ecosystems (Wang et al., 2025), backdoors can propagate beyond a single instance; we further empirically show they persist across federated rounds and generalize.

To address backdoor integrity under secure aggregation, we introduce INDEXGUARD, an unsupervised *index-only* pre-aggregation vetting mechanism for federated PEFT. We target self-attention because PEFT parameter updates on attention projections exhibit *concentrated salience*: importance is highly skewed across heads/projections and many heads are redundant (Michel et al., 2019; Voita et al., 2019), while fine-tuning shifts which projections become salient for the downstream objective (Zhao & Bethard, 2020). Prior work shows $Q/V$ adapter updates are especially discriminative for PEFT backdoors (Sun et al., 2025), consistent with backdoor objectives systematically manipulating attention patterns (Lyu et al., 2023). Our key insight is that *shared objectives induce shared salience supports*: within one FL run (fixed backbone/adapter placement), benign clients concentrate salience on a common task-driven subset of attention coordinates, whereas poisoned clients also emphasize a trigger-specific subset, inducing high within-poison and low benign–poison Top-$K$ index overlap even under non-IID data (as we show empirically).

To expose this structure without revealing values, each client maps its local adapter change into a fixed, shared *PEFT parameter space*. Coordinate IDs are defined over the monitored trainable PEFT parameters so they are consistent across clients and rounds (see §5.1). For PEFT variants with different parameterizations, clients first map their trainable updates into the same monitored sketch space (i.e., the shared $Q/V$ adapter-parameter coordinates) to preserve consistent coordinate IDs across clients. Each client sends only an *index sketch* $S_i$ containing the Top-$K$ salient coordinate IDs ($K \ll M$, where $M$ is the number of scalar coordinates in PEFT parameter space). The server separates clients via Index-Overlap Similarity $\text{IOS}_{ij} = |S_i \cap S_j|/K$, selects a vetted cohort, and then runs secure aggregation over only that cohort's real-valued adapter updates (Bonawitz et al., 2017). Unlike sign/low-bit sketches that communicate value-bearing signals and may reduce accuracy under quantization (Bernstein et al., 2018; Alistarh et al., 2017), INDEXGUARD matches practical secure aggregation and is compatible with optional HE and adapter-level DP; residual leakage through indices and mitigations are analyzed in Appx. B.

**Our Contributions.** To the best of our knowledge, IN-DEXGUARD is the first *PEFT-structure-aware, index-only pre-aggregation vetting* method for *federated* PEFT that avoids access to per-client real-valued updates and satisfies secure aggregation constraints. We provide theory for index-based vetting in federated PEFT, showing that Top-$K$ overlap in the monitored *self-attention $Q/V$ PEFT parameter-update space* exhibits alignment, stability, and scale-robustness under mild salience-concentration assumptions across heads/projections (see §6). Extensive experiments under standard threat models across adapters (LoRA, AdaLoRA, QLoRA, LoRA+, DoRA), attacks (InsertSent, RIPPLES, Syntactic, StyleBkd), LLMs models (Llama-2-7B, Llama-3.2-3B, Llama-3-8B and BERT/RoBERTa), and datasets (AG News, SST2, and instruction-following settings) show that INDEXGUARD sharply separates benign from backdoored clients, often from early FL rounds.

**Conflict of Interest Disclosure.** The authors declare no financial conflicts of interest related to this work.

## 2. Background and Motivation

### 2.1. Preliminaries

**PEFT for LLMs.** PEFT adapts a pretrained Transformer by freezing backbone weights $W_0$ and training a small parameter set $\theta$ (e.g., adapters/prefix/prompt), reducing optimization cost and, in federated settings, per-round communication by exchanging only $\theta$ (Houlsby et al., 2019; Hu et al., 2022). For LoRA, the method targets a set of projection matrices $\mathcal{P}$ (typically within self-attention); each targeted projection $W_{\text{base}} \in \mathbb{R}^{d_{\text{out}} \times d_{\text{in}}}$ is updated via a low-rank residual (Hu et al., 2022; Vaswani et al., 2017):

$$W = W_{\text{base}} + \Delta W, \qquad \Delta W = \frac{\alpha}{r} BA, \qquad (1)$$

where $A \in \mathbb{R}^{r \times d_{\text{in}}}$, $B \in \mathbb{R}^{d_{\text{out}} \times r}$, and $r \ll \min(d_{\text{in}}, d_{\text{out}})$. The trainable state is $\theta = \{A, B\}$ (and possibly per-layer scalings), while $W_0$ remains fixed.

**PEFT variants.** SLoRA variants keep a frozen backbone but change the update/budget: QLoRA trains LoRA through a 4-bit quantized backbone to cut memory (Dettmers et al., 2023); AdaLoRA allocates rank across layers under a budget (Zhang et al., 2023); LoRA+ uses separate learning rates/scales for the two factors (Hayou et al., 2024); and DoRA adapts the directional component of a magnitude–direction decomposition without extra inference cost after merging (Liu et al., 2024). We view all as inducing an effective projection update $\Delta W$, enabling a unified induced-update representation for analysis and comparison.

**Federated PEFT protocol and aggregation.** We consider cross-device/cross-silo federated PEFT over rounds

$t = 0, \ldots, T - 1$, where a frozen backbone $W_0$ is shared and only lightweight adapters are trained. The server maintains global adapter $\theta^{(t)}$ at round $t$. Sampled clients set $\mathcal{C}_t$ downloads $\theta^{(t)}$, fine-tune locally on private data with the backbone frozen, to obtain updated adapter parameters $\theta^{(t+1)}$ (McMahan et al., 2017; Zhang et al., 2024b). Servers often FedAvg adapter parameters, but for LoRA, this is fragile because the $BA$ factorization is non-unique and breaks under heterogeneous ranks $r_i$ (Wang et al., 2024). A rank-robust alternative aggregates in effective-update space by mapping clients to their induced update $\Delta W_i$ and averaging:

$$\Delta W^{(t+1)} = \sum_{i \in \mathcal{C}_t} \frac{n_i}{\sum_{j \in \mathcal{C}_t} n_j} \Delta W_i^{(t+1)},$$
$$\Delta W_i^{(t+1)} = \frac{\alpha}{r_i} B_i^{(t+1)} A_i^{(t+1)}. \tag{2}$$

where $n_i$ is client $i$'s sample count. Because averaging $\Delta W^{(t+1)}$ can increase rank (up to $\sum_{i \in \mathcal{C}_t} r_i$), implementations refactorize/compress the aggregate $\Delta W^{(t+1)}$ to a target rank to form the next-round adapter state (Wang et al., 2024). This is approximate: $\theta^{(t+1)}$ is chosen so that its induced update $\Delta W(\theta^{(t+1)})$ best matches $\Delta W^{(t+1)}$ under the rank/budget constraint. Within a run, all participating clients use the same PEFT configuration for aggregation.

**Backdoors in adapter-based LLMs and implications for federated PEFT.** A backdoor is a training-time integrity compromise: the adapted model behaves normally on clean inputs but follows attacker-chosen behavior under a hidden trigger. In adapter-based customization, the attacker perturbs only the adapter state $\theta$ (e.g., LoRA factors) so the induced update $\Delta W(\theta)$ encodes a trigger–behavior association while the backbone $W_0$ remains frozen. Textual triggers range from explicit token/phrase/sentence insertion to implicit syntactic or style transformations in classification, and prompt-embedded cues in instruction following/generation that elicit targeted behaviors (e.g., toxic completions) with little clean-utility loss. Triggers can look benign (e.g., a year string like "2024") yet reliably activate harmful behavior and persist under further fine-tuning, motivating integrity defenses beyond privacy (Hubinger et al., 2024). Adapters also reshape the attack surface: confining optimization to a small structured subspace (e.g., low-rank attention updates) enables high-impact manipulation with a compact footprint and weak utility-based detection signals.

Let $\mathcal{D} = \mathcal{D}_c \cup \mathcal{D}_b$ be the empirical fine-tuning distribution, where $\mathcal{D}_c$ contains clean pairs $(x, y)$ and $\mathcal{D}_b$ contains backdoor pairs $(x^\tau, y^\star)$ formed by applying a trigger operator $\mathcal{T}\tau$ (token/sentence/style/syntax) to obtain $x^\tau = \mathcal{T}\tau(x)$ with attacker-chosen target $y^\star$. With frozen backbone $W_0$ and trainable adapters $\theta$, the poisoned SFT objective (token-

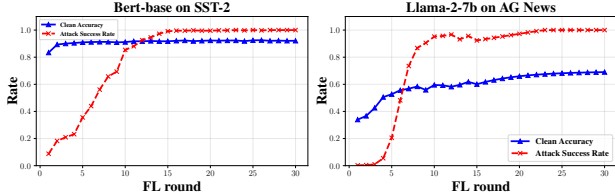

*Figure 1.* Backdoor propagation in federated PEFT under low-budget poisoning (Llama-2-7B on AG News).
level cross-entropy/negative log-likelihood $\mathcal{L}$) is:

$$\theta^\star = \arg\min_\theta \; \mathbb{E}_{(x,y) \sim \mathcal{D}_c} \big[ \mathcal{L}\big(f_{W_0,\theta}(x), y\big) \big]$$
$$+ \lambda \, \mathbb{E}_{(x^\tau, y^\star) \sim \mathcal{D}_b} \big[ \mathcal{L}\big(f_{W_0,\theta}(x^\tau), y^\star\big) \big], \tag{3}$$

where $\lambda$ controls the clean–backdoor tradeoff (i.e., poisoning strength), balancing nominal behavior on clean inputs against attacker-chosen behavior on triggered inputs. In federated PEFT, data are partitioned across clients, motivating analysis of how this objective interacts with federated optimization/aggregation under privacy constraints.

## 2.2. Motivation

We motivate INDEXGUARD by showing that triggers can propagate and accumulate across rounds in federated PEFT. We consider $N=10$ clients with only 2 attackers ($f=20\%$) and per-attacker poisoning rate $\rho=0.05$, and run the INSERT-SENT backdoor attack. Figure 1 reports Clean Accuracy (CA) and Attack Success Rate (ASR) over FL rounds for two representative tasks (BERT-base on SST-2 and Llama-2-7B on AG News). Despite the small poisoned mass each round, ASR rises quickly and approaches 1.0 as aggregation repeats, evidencing cumulative backdoor amplification through iterative mixing of PEFT updates. Crucially, CA remains largely stable and converges, yielding a high-utility / high-compromise regime where standard validation fails to reveal the attack. Overall, it exposes a core systems risk: without FL-compatible attack detection and client vetting, backdoors can reach near-certain trigger success while preserving clean performance, motivating lightweight defenses that can flag poisoned contributions early.

## 3. Related Work

**Centralized backdoor detection in NLP/LLMs.** A substantial body of work studies *centralized* backdoor detection/mitigation assumes full access to a trained model often with clean data or inference queries. Representative centralized approaches include trigger reverse-engineering (e.g., Neural Cleanse (Wang et al., 2019)), statistical spectral outlier detection (Tran et al., 2018), and input-time defenses for textual backdoor such as ONION (Qi et al., 2021b). Several NLP-focused centralized backdoor detectors have also been proposed: T-Miner (Azizi et al., 2021) synthesizes probe texts via a seq2seq generator to surface Trojan

triggers in text classifiers; AttenTD (Lyu et al., 2022) flags Trojaned BERTs by detecting attention-focus drifting; PIC-COLO (Liu et al., 2022) exposes complex backdoors in Transformer NLP models via trigger-oriented analysis; and MNTD (Xu et al., 2021) detects AI Trojans using a framework trained over shadow models. These methods cannot attribute backdoors to individual clients in FL.

**Centralized backdoor detection for PEFT/adapters.** Recent work targets backdoors *inside PEFT modules* rather than full-model weights. PEFTGuard (Sun et al., 2025) trains a *supervised* meta-classifier on a labeled corpus, PAD-Bench, achieving strong detection and transfer across PEFT variants/ranks. However, it trains a labeled corpus that is not available in the real-world fast-evolving LLMs.

**Backdoor defenses for conventional FL (full-model updates).** Defenses for standard FL primarily rely on robust aggregation or update filtering, assuming server access to *per-client real-valued* updates. Classic Byzantine-robust rules (e.g., Krum/Multi-Krum (Blanchard et al., 2017), and coordinate-wise median/trimmed mean (Yin et al., 2018)) downweight or exclude anomalous clients based on update geometry. Backdoor-specific methods further apply clustering with clipping or noise (e.g., FLAME (Nguyen et al., 2022)), frequency-domain filtering (e.g., FreqFed (Fereidooni et al., 2024)), or direction-alignment checks (AlignIns (Xu et al., 2025)). These approaches rely on *update values*, incompatible with secure aggregation, and can degrade under non-IID heterogeneity.

**Backdoor defenses for federated PEFT.** Existing FL backdoor defenses target standard weight/gradient updates and rely on *per-client real-valued updates* (or server-side access to them) to compute robust statistics, or outlier scores. This gap motivates INDEXGUARD: an *index-only, pre-aggregation* vetting signal for federated PEFT that enables the server to form a vetted cohort *without* access to per-client update values, compatible with secure aggregation. INDEX-GUARD re-designs this for *secure federated PEFT backdoor vetting* by using adapter-salience indices with backdoor-tailored Top-$K$ calibration, adaptive privacy mechanism, and a cohesion/outlier filter to separate benign vs. malicious clients *before* aggregation. Table 1 summarizes the capabilities of prior backdoor defenses.

## 4. Threat Model

**Setting.** FL with a coordinating server; clients fine-tune PEFT adapters (LoRA variants) on their private data $\mathcal{D}$.

**Poisoned contributors.** An adversary $\mathcal{A}$ corrupting a fraction $f$ of $N$ clients ($|\mathcal{A}| = fN$) carries out backdoor-inducing updates by intentionally optimizing a mixed objective over clean and triggered samples, so as to enforce $\tau \mapsto y^\star$. Unless otherwise stated, we focus on the *shared-trigger* regime, where corrupted clients use similar triggers and target behavior. This is the attacker-favorable set-

*Table 1.* **Backdoor defense comparison**. **Unsup**: unsupervised; **NonIID**: Evaluated under non-IID; **NoUpd**: no per-client real-valued updates; **SecAgg**: compatible with secure aggregation; **NoMerge**: no backbone instantiation/inference, − = not applicable

| Method | Unsup | NonIID | NoUpd | SecAgg | NoMerge |
|---|---|---|---|---|---|
| Neural Cleanse | ✓ | ✗ | ✓ | – | ✗ |
| Spectral Signatures | ✓ | ✗ | ✓ | – | ✗ |
| ONION / T-Miner | ✓ | ✗ | ✓ | – | ✗ |
| AttenTD / PICCOLO / MNTD | ✓ | ✗ | ✓ | – | ✗ |
| PEFTGuard | ✗ | ✗ | ✗ | ✗ | ✓ |
| Krum/Multi-Krum | ✓ | ✗ | ✗ | ✗ | ✓ |
| Median/TrimmedMean | ✓ | ✗ | ✗ | ✗ | ✓ |
| FLAME / FreqFed / AlignIns | ✓ | ✓ | ✗ | ✗ | ✓ |
| INDEXGUARD (ours) | ✓ | ✓ | ✓ | ✓ | ✓ |

ting, since aligned poisoned objectives reinforce each other through repeated aggregation and lead to faster propagation. The remaining clients, denoted by $\mathcal{B}$, are benign.

**Beyond shared triggers.** We also evaluate mixed-trigger attacks as a stress test, where corrupted clients are split across different attack families and may not form a single cohesive malicious cluster. This setting is not our main threat model because heterogeneous triggers dilute the attack signal, but Appendix F.6 shows a lightweight mixed-trigger-aware extension of INDEXGUARD. We also evaluate dishonest reported-support manipulation as an adaptive extension; the core protocol assumes clients report Top-$K$ sketches derived from their local updates, while protected reporting can enforce this in stricter deployments.

**Server Corruption.** We consider an adversary that corrupts the server attempting to infer information about private local models. Since our approach adopts secure aggregation (Mansouri et al., 2023) protecting the local updates of the clients from the corrupted server, the server sees only index sets and the aggregated result.

### 4.1. Why We Focus on the Shared-Trigger Regime

Our main threat model focuses on the shared-trigger regime, where malicious clients coordinate on the same trigger and target behavior. This choice is attacker-favorable: when poisoned clients optimize the same backdoor objective, their updates reinforce each other during repeated federated aggregation, making the backdoor propagate quickly while clean accuracy remains high. This is also the regime where malicious clients are expected to induce a coherent salience footprint, which is the primary signal used by IN-DEXGUARD. This assumption is already part of our threat model, where corrupted clients are assumed to use similar triggers across clients. Figure 2 empirically supports this choice in a 10-client setting with 30% malicious clients (3 attackers) and poison rate $\rho = 0.1$. When all attackers use InsertSent, ASR increases rapidly from 0.167 in round 1 to 0.829 by round 5 and nearly 0.99 by round 9, while clean accuracy stays around 0.91. In contrast, when the same three attackers are split evenly across InsertSent, RIPPLES,

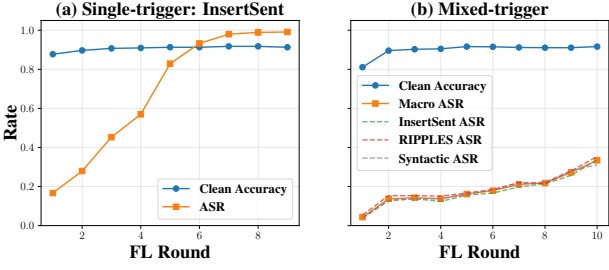

*Figure 2.* Shared-trigger poisoning is more attacker-favorable than mixed-trigger poisoning. We use $N = 10$ clients, $f = 30\%$ malicious clients, and poison rate $\rho = 0.1$. In the shared-trigger case, all three attackers use InsertSent; in the mixed-trigger case, one attacker uses InsertSent, one uses RIPPLES, and one uses Syntactic.

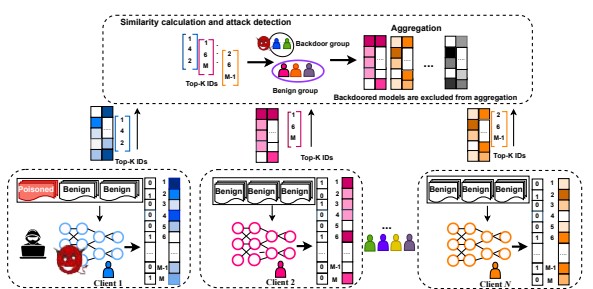

*Figure 3.* **INDEXGUARD workflow:** clients send Top-$K$ salience indices; the server clusters by IOS to vet clients before aggregation.

and Syntactic triggers, the macro-ASR grows much more slowly and reaches only $0.334$ by round 10. Thus, mixed-trigger poisoning is less favorable for the attacker because heterogeneous malicious objectives are diluted by aggregation. We therefore use the shared-trigger regime as the main evaluation setting and consider mixed-trigger attacks as a stronger heterogeneity stress test.

## 5. INDEXGUARD

Many FL defenses assume access to unshielded per-client real-valued updates or rely on supervised detectors trained on a labeled server-side corpus. In contrast, INDEX-GUARD (Fig. 3) is a *pre-aggregation* integrity filter for federated PEFT that detects backdoored clients using an index-only signature of Top-$K$ index sets in a fixed monitored coordinate space $\theta_P$ (LoRA on self-attention $Q/V$ only), where $M := |\theta_P|$ is the number of scalar trainable PEFT coordinates. This signature exhibits higher mutual overlap among malicious clients than between malicious and benign clients. INDEXGUARD computes Top-$K$ in the *canonical* PEFT parameterization broadcast each FL round: clients start from the same $\theta_{\mathcal{P}}^{(t)}$ and compute $\Delta\theta_i^{(t)}$, so coordinate IDs (parameter-name order over trainable adapter tensors) are comparable across clients and rounds. While some PEFT forms admit equivalent reparameterizations (e.g., LoRA $\Delta W = BA = (BR)(R^{-1}A)$), INDEX-

GUARD operates on the deployed adapter tensors distributed by the FL protocol. We determine $K$ once in a short warm-up and keep it fixed thereafter. INDEXGUARD cleanly separates *index-only vetting* from *secure aggregation* as described by Algorithm 1: **Phase A (index-only):** clients send $S_i$; the server computes $\mathcal{V}_t$ and announces the vetted cohort. **Phase B (privacy-preserving aggregation):** After phase A, only clients in $\mathcal{V}_t$ participate in secure aggregation of their updates. As a result, INDEXGUARD enables (i) secure-aggregation–compatible pre-screening, (ii) low bandwidth overhead, and (iii) generalization across backbones and PEFT variants without backdoor corpora.

---

**Algorithm 1** INDEXGUARD, one FL round

---

**Input:** Frozen backbone $W_0$, initial adapters $\theta^{(0)}$, targeted $\mathcal{P}$ (self-attention $Q/V$), sketch budget $K$, number of clusters $L$, threshold $\gamma$

**Phase A: Index-only vetting**

**for all** $i \in \mathcal{C}_t$ **in parallel do**

$\quad \theta_i^{(t+1)} \leftarrow \text{LocalTrain}(W_0, \theta^{(t)}; \mathcal{D}_i)$

$\quad \Delta\theta_{i,\mathcal{P}}^{(t)} \leftarrow \theta_{i,\mathcal{P}}^{(t+1)} - \theta_{\mathcal{P}}^{(t)}, u_i^{(t)} \leftarrow \vec{(}\Delta\theta_{i,\mathcal{P}}^{(t)})$

$\quad s_i^{(t)} \leftarrow |u_i^{(t)}|, S_i \leftarrow \text{TopKIdx}(s_i^{(t)}, K)$

$\quad$ Send $S_i$ to server

**end for**

Server computes $\text{IOS}_{ij}$ and clusters $\{S_i\}_{i \in \mathcal{C}_t}$ into $L$ clusters $\{C_\ell\}_{\ell=1}^L$

Compute $\text{Coh}(C_\ell)$ for each cluster; let $\ell^\star \leftarrow \arg\max_\ell |C_\ell|$

$\mathcal{C}_t^{\text{sus}} := \bigcup_{\ell: C_\ell \text{ is flagged suspicious}} C_\ell$

$\mathcal{V}_t \leftarrow \mathcal{C}_t \setminus \mathcal{C}_t^{\text{sus}}$

broadcast $\mathcal{V}_t$

**Phase B: Private aggregation**

**for all** $i \in \mathcal{V}_t$ **in parallel do**

$\quad$ Participate in PrivAgg with update $\Delta\theta_i^{(t)}$

**end for**

$\theta_{\mathcal{P}}^{(t+1)} \leftarrow \theta_{\mathcal{P}}^{(t)} + \text{PrivAgg}\left(\{\Delta\theta_{i,\mathcal{P}}^{(t)}\}_{i \in \mathcal{V}_t}\right)$

---

### 5.1. Client-side: Local Training/ signature Construction

**Step 1: Local PEFT update.** Client $i$ downloads $\theta^{(t)}$ and runs local PEFT with frozen backbone $W_0$ to obtain $\theta_i^{(t+1)}$. For a targeted projection $W_{\text{base}} \in \mathbb{R}^{d_{\text{out}} \times d_{\text{in}}}$, let $R(\theta)$ denote the adapter residual parameterized by the PEFT module; for LoRA/QLoRA with rank $r$, $R(\theta) = \frac{\alpha}{r}BA$. The resulting dense residual update in that projection is $\delta W_i^{(t)} := R(\theta_i^{(t+1)}) - R(\theta^{(t)})$.

**Step 2: Shared PEFT coordinate space.** Let $\theta_{\mathcal{P}}$ denote the monitored *trainable* PEFT parameters (LoRA on $Q/V$ only), with $M := |\theta_{\mathcal{P}}|$ and fixed global coordinate IDs (parameter-name order shared across clients). For other LoRA-family variants (LoRA+/AdaLoRA/DoRA), we define $\theta_{\mathcal{P}}$ as the *trainable* tensors attached to the same Q/V adapter modules (after the method's internal reparameter-

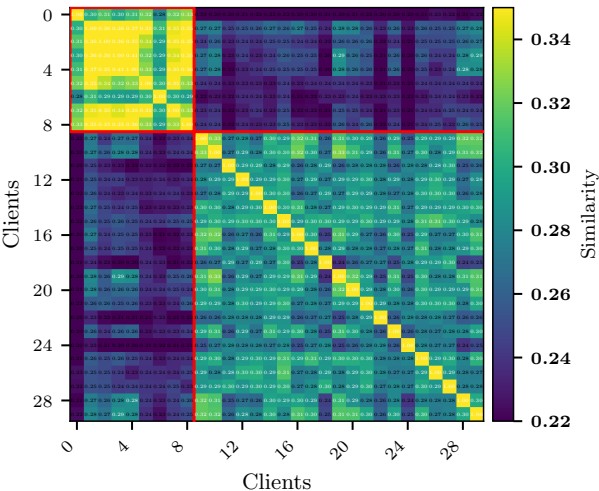

*Figure 4.* Pairwise IOS heatmap for $N = 30$ for Llama-3.2-3B on AG News under the StyleBkd attack. Clients 1–9 are attackers.

ization) and flatten them in a fixed parameter-name order, so each reported index always refers to a unique canonical coordinate. Client $i$ forms the adapter delta w.r.t. the broadcast model: $\Delta\theta_i^{(t)} := \theta_{i,\mathcal{P}}^{(t+1)} - \theta_{\mathcal{P}}^{(t)}$ and flattens it as $u_i^{(t)} := \text{vec}(\Delta\theta_i^{(t)}) \in \mathbb{R}^M$.

**Step 3: Salience and index sketch.** We set salience entry-wise as $s_i^{(t)}[m] := |u_i^{(t)}[m]|$, equivalently, $s_i^{(t)} = |\text{vec}(\Delta\theta_i^{(t)})|$ over the monitored LoRA $Q/V$ parameters. Client $i$ sends only $S_i^{(t)} = \text{TopKIdx}(s_i^{(t)}, K)$ to the server. In practice, we compute TopKIdx with standard library partial selection (e.g., `torch.topk`/`argpartition`). Although triggers are textual, the backdoor objective enforces a consistent conditional mapping ($\to y^\star$), inducing a low-entropy update signal that concentrates on a stable subset of $Q/V$ LoRA coordinates. Benign updates, by contrast, spread salience across diverse task semantics and client heterogeneity, yielding lower within-benign than within-adversary Top-$K$ overlap under a shared trigger.

## 5.2. Server-side: IOS Similarity, Clustering, and Vetting

**Step 4: Index-Overlap Similarity (IOS).** Given $\{S_i\}_{i \in \mathcal{C}_t}$, the server computes pairwise similarity:

$$\text{IOS}_{ij} = \frac{|S_i \cap S_j|}{K}, \qquad i, j \in \mathcal{C}_t, \qquad (4)$$

which measures the discrete overlap of indices across salient support patterns without using real-valued updates. **Remark.** While the server computes IOS, clients can reduce semantic leakage from $S_i$ by transmitting a protected representation. In particular, clients may transmit *salted tokens* $\tilde{S}_i^{(t)} = \{ \text{Hash}(S_{i,j}^{(t)} \| \text{salt}_t) : S_{i,j}^{(t)} \in S_i^{(t)} \}$, where $\text{salt}_t$ is a round-specific salt shared among clients. Since IOS depends only on set intersection, salted tokens preserve detection accuracy while making the reported coordinates opaque to

the server. While salting protects $S_i$, it introduces two integrity gaps. First, a corrupted client can submit an arbitrary value $\tilde{S}_i^{(t)}$ that is not derived from its local model (i.e., $S_i^{(t)}$ is *inconsistent* with the claimed computation). Second, even if client $i$ reports a correctly formed $\tilde{S}_i^{(t)}$, it can still submit model parameters $\theta_i$ trained on poisoned local data $\tilde{\mathcal{D}}_i$ to the private aggregation, thereby injecting a backdoor. To address these challenges, we adopt zero-knowledge proofs, which allow the server to verify the relevant computations without revealing $S_i$ or $\theta_i$ (see Appx. E for details).

**Step 5: Clustering and suspicious cluster selection.** The server forms the similarity matrix $[\text{IOS}_{ij}]$ and clusters clients using a similarity-based method (e.g., hierarchical clustering on $1 - \text{IOS}_{ij}$), yielding $L$ clusters $\{C_\ell\}_{\ell=1}^L$. For any cluster $C_\ell$ with $|C_\ell| \geq 2$, we define its *cohesion* as the average intra-cluster similarity,

$$\text{Coh}(C_\ell) = \frac{1}{|C_\ell|(|C_\ell| - 1)} \sum_{\substack{i,j \in C_\ell \\ i \neq j}} \text{IOS}_{ij}.$$

Let $\ell^\star := \arg\max_\ell |C_\ell|$ denote the largest benign-majority cluster. A cluster $C_\ell$ is flagged as suspicious if it is a *non-majority* cohesion outlier, i.e.,

$$\text{Coh}(C_\ell) \geq \text{median}_{\ell'=1}^L \{\text{Coh}(C_{\ell'})\} + \gamma \quad \text{and} \quad \ell \neq \ell^\star,$$

optionally with a conservative size cap $|C_\ell| \leq \beta |C_{\ell^\star}|$ ($\beta < 1$) to avoid vetoing large secondary benign modes under benign multi-modality. We then set

$$\mathcal{C}_t^{\text{sus}} := \bigcup_{\ell: C_\ell \text{ is flagged suspicious}} C_\ell, \qquad \mathcal{V}_t := \mathcal{C}_t \setminus \mathcal{C}_t^{\text{sus}}.$$

In the shared-trigger regime, malicious clients exhibit low-noise, trigger-aligned salience supports, yielding unusually high within-cluster IOS and high Coh, whereas benign clients exhibit heterogeneous, task-driven salience supports. As shown in Fig. 4, *within-group* IOS similarity (attacker–attacker and benign–benign) is systematically higher than *cross-group* similarity (attacker–benign), producing a clear cohort separation from Top-$K$ overlap alone. Moreover, the attacker block is more cohesive: since the attackers optimize the same objective, their updates concentrate salience on shared trigger-aligned Top-$K$ indices, yielding higher mutual overlap than the heterogeneous benign population.

## 5.3. Selecting the Top-$K$ budget

Rather than fixing $K$ ad hoc, we set it via a *coverage+stability* rule tailored to PEFT on LLMs. We set $K$ once before training and keep it fixed for all rounds. For each client $i$, we compute a nonnegative salience/importance vector $s_i$ over the monitored $Q/V$ PEFT coordinate space (with fixed global IDs). After sorting $s_i$ in descending

order, let $s_{i,j}$ be the $j$-th largest entry and define cumulative coverage $C_i(K) = \frac{\sum_{j=1}^{K} s_{i,j}}{\sum_{j=1}^{M} s_{i,j}}$, which is monotone in $K$, so the smallest $K_i^{\text{cov}} \leq K_{\max}$ with $C_i(K_i^{\text{cov}}) \geq \tau$ is found by binary search. We then set $K_i^\star$ as the smallest $K \in [K_i^{\text{cov}}, K_{\max}]$ satisfying the resample overlap $\rho_i(K) \geq \rho_0$ (mean pairwise Top-$K$ overlap; Appx. A). Finally, the server fixes a global $K = \text{clip}(\text{median}_i K_i^\star, K_{\max})$ and each client transmits exactly $K$ indices (details in Appx. A).

### 5.4. Time and bandwidth overhead

Phase A adds only linear-time instrumentation: one pass to compute importance scores and one-time linear-time Top-$K$ selection, i.e., $T_A^{(t)} = O(M)$ (is amortized). Phase A communicates only the Top-$K$ index set, $B_A^{(t)} = K\, b_{\text{idx}}$ bits, so relative overhead over the secure-aggregation payload $B_{\text{agg}}^{(t)}$ is $\frac{K\, b_{\text{idx}}}{M\, b_{\text{val}}}$, negligible when $K \ll M$; the added uplink time is $\Delta T_{\text{comm}}^{(t)} \approx \frac{K\, b_{\text{idx}}}{R}$. See Appx. C for details.

## 6. Theoretical Analysis

We justify Index-Overlap Similarity (IOS) as a similarity primitive for index-only vetting. Each client reports $S_i = \text{TopK}(s_i) \subseteq [M]$ where $s_i := |\Delta\theta_{i,\mathcal{P}}| \in \mathbb{R}_{\geq 0}^M$ is the nonnegative salience vector in the canonical PEFT coordinate space, and $IOS(s_i, s_j) := |S_i \cap S_j|/K$. Space constraints limit us to a compact summary here; full statements and proof sketches are in Appx. D.

### 6.1. IOS foundations: alignment, stability, robustness

**Alignment.** If $IOS(s, t) \geq \rho$, then the overlap set carries nontrivial salience mass under mild regularity of Top-$K$ magnitudes. Concretely, if the Top-$K$ entries of $s$ lie in $[\underline{s}, \overline{s}]$ with $\kappa_s = \overline{s}/\underline{s}$ (and similarly $\kappa_t$), then $sum_{j \in S_s \cap S_t} s_j \geq \frac{\rho}{\kappa_s} \sum_{j \in S_s} s_j$ and $\sum_{j \in S_s \cap S_t} t_j \geq \frac{\rho}{\kappa_t} \sum_{j \in S_t} t_j$. Thus, IOS lower-bounds the fraction of Top-$K$ salience mass shared by two clients (Appx. D).

**Stability.** Top-$K$ supports are reproducible when the $K$-boundary margin dominates noise. Let $\hat{s} = s + \varepsilon$ where $\varepsilon_j$ are independent sub-Gaussian noises with scale $\sigma$ (e.g., SGD/DP noise, reduced by EMA smoothing). With Top-$K$ margin $\Delta := s_{(K)} - s_{(K+1)}$, we have $\Pr[\text{TopK}(\hat{s}) \neq \text{TopK}(s)] \leq 2M \exp\left(-\frac{\Delta^2}{8\sigma^2}\right)$, and $\text{TopK}(\hat{s}) = \text{TopK}(s)$ whenever $\max_j |\varepsilon_j| \leq \Delta/2$ (Appx. D).

**Robustness.** Since IOS depends only on rank order, it is invariant to global scaling and resilient to mild diagonal preconditioning. If $\tilde{s} = Ds$ with $D = \text{diag}(d_j)$ and $d_j \in [1 - \delta, 1 + \delta]$, then $\text{TopK}(\tilde{s}) = \text{TopK}(s)$ provided $\Delta > 2\delta\, s_{(K)}$; more generally, only coordinates within an $O(\delta)$ band around the threshold can flip (Appx. D).

### 6.2. Separation under a similar backdoor imprint.

Let $\mathcal{B}$ be benign clients (task loss) and $\mathcal{A}$ adversaries (task+backdoor) sharing a trigger. Define group-wise expected IOS: $IOS_{\mathcal{B}\mathcal{B}}, IOS_{\mathcal{A}\mathcal{A}}, IOS_{\mathcal{B}\mathcal{A}}$. Under non-IID data, benign Top-$K$ supports differ across clients; model this by the mean (noise-free) overlap $\rho_{\mathcal{B}} := \mathbb{E}\left[\frac{|S_i^\star \cap S_j^\star|}{K} \mid i \neq j, i, j \in \mathcal{B}\right]$, where $S_i^\star := \text{TopK}(\mu_i)$ is the Top-$K$ support of the stationary mean $\mu_i$ in $s_i = \mu_i + \xi_i$. We assume a *trigger-imprint set* $T \subseteq [M]$ with $|T| = K_T \leq K$ such that adversaries include $T$ in their reported Top-$K$ with high probability while benign clients overlap weakly with $T$ on average (parameter $\eta$); and let $\delta_{\mathcal{A}}, \delta_{\mathcal{B}}$ capture Top-$K$ instability. Then (Appx. D) we obtain **i)** $IOS_{\mathcal{A}\mathcal{A}} \geq (1 - 2\delta_{\mathcal{A}})\frac{K_T}{K}$, **ii)** $IOS_{\mathcal{B}\mathcal{A}} \leq \rho_{\mathcal{B}\mathcal{A}} + \eta + \delta_{\mathcal{A}}$, **iii)** $IOS_{\mathcal{B}\mathcal{B}} \leq \rho_{\mathcal{B}} + 2\delta_{\mathcal{B}}$, where $\rho_{\mathcal{B}\mathcal{A}}$ is the noise-free task-driven benign–adversary Top-$K$ overlap. Hence, whenever $(1 - 2\delta_{\mathcal{A}})\frac{K_T}{K} > \rho_{\mathcal{B}} + 2\delta_{\mathcal{B}}$, adversaries form a more cohesive IOS subpopulation than benign clients, while cross-group similarity remains controlled—the regime exploited by clustering on $1 - IOS$.

## 7. Experimental Analysis

### 7.1. Experimental Setup

**Backbones and PEFT.** We evaluate both encoder-only backbones (BERT-base, RoBERTa-base) and decoder-only LLMs (Llama-2-7B, Llama-3-3B) under LoRA-family PEFT (LoRA, QLoRA, LoRA+, AdaLoRA, DoRA). Adapters are inserted into self-attention projections (Q/V for decoder-only; query/value for encoder-only). For all PEFT variants, INDEXGUARD sketches are formed in the *canonical monitored* Q/V coordinate space so Top-$K$ indices are comparable across clients and rounds.

**Tasks, attacks, and non-IID partitions.** We use SST-2, AG News, and instruction-following backdoor corpora `toxic-backdoors-alpaca` and `toxic-backdoors-hard` (derived from Alpaca). We instantiate four standard textual backdoors: rare-word triggers (RIPPLES), InsertSent, syntactic triggers via SCPN, and StyleBkd. Unless stated otherwise, each malicious client poisons a $\rho$=5% fraction of its local samples. Client data heterogeneity is modeled via Dirichlet label partitions $\text{Dir}(\alpha)$; we report $\alpha$=0.3 by default.

**Federated setting and reporting.** Unless otherwise specified, detection experiments use $N$=30 clients with $|A|$=6 malicious clients. We report: (i) *attack success rate* (ASR), the fraction of triggered test inputs mapped to the attacker target $y^\star$; and (ii) *AUC* for client vetting, obtained by sweeping the INDEXGUARD suspiciousness score threshold to trace the TPR–FPR tradeoff. Full hyperparameters and ablations are provided in Appx. F. The code is publicly available at (IndexGuard).

*Table 2.* IndexGuard detection across tasks and backdoors

| Dataset | Attack | INDEXGUARD | |
|---|---|---|---|
| | | Acc (%) | AUC |
| toxic-backdoors-alpaca | Word | $100.0 \pm 0.00$ | $1.000 \pm 0.000$ |
| toxic-backdoors-hard | Sentence | $100.0 \pm 0.00$ | $1.000 \pm 0.000$ |
| AG News | InsertSent | $100.0 \pm 0.00$ | $1.000 \pm 0.000$ |
| | RIPPLES | $98.2 \pm 0.31$ | $0.996 \pm 0.005$ |
| | Syntactic | $99.3 \pm 0.00$ | $0.998 \pm 0.002$ |
| | StyleBkd | $100.0 \pm 0.00$ | $1.000 \pm 0.000$ |
| SST | InsertSent | $100.0 \pm 0.00$ | $1.000 \pm 0.000$ |
| | RIPPLES | $99.6 \pm 0.16$ | $0.996 \pm 0.003$ |
| | Syntactic | $100.0 \pm 0.00$ | $1.000 \pm 0.000$ |
| | StyleBkd | $99.3 \pm 0.26$ | $0.982 \pm 0.008$ |

## 7.2. Performance Evaluation

**Detection effectiveness across tasks and triggers.** Table 2 reports INDEXGUARD's client-vetting performance across heterogeneous NLP tasks and backdoor mechanisms. Across instruction-following corpora (TB-Alp/Hard) and downstream benchmarks (AG News, SST), INDEX-GUARD attains near-ceiling detection, with *perfect separation* (Acc= 100%, AUC= 1.0) in most settings. The only noticeable drops occur for stealthier triggers—rare-word (RIPPLES) and style-based attacks—where AUC remains high (down to 0.982 on SST StyleBkd), and accuracy decreases slightly (to $\approx 98\%$ on AG News RIPPLES). The very small standard deviations indicate stable ranking and threshold robustness. Overall, index-only salience sketches provide a strong, attack-agnostic signal for detecting poisoned clients across diverse tasks/triggers, making INDEX-GUARD an accurate front-end vetting step pre-aggregation. **Comparison to prior backdoor detectors and applicability to federated PEFT.** To the best of our knowledge, there is no existing *federated* PEFT/LoRA backdoor detector that is simultaneously (i) unsupervised, (ii) compatible with secure aggregation, and (iii) value-free (i.e., does not require per-client real-valued updates). We therefore compare to PEFTGuard and the closest *centralized* PEFT detectors, which vet adapters from real-valued weights or via supervised/meta-trained detectors learned from labeled benign/poisoned adapters. Table 3 summarizes detection quality by averaging across two evaluation regimes: **SC** (standard supervised NLP tasks) and **IF** (instruction-following toxic backdoor corpora). INDEXGUARD attains near-ceiling AUC in both regimes (SC: $\approx 0.998$, IF: 1.0) with correspondingly high detection accuracy (SC: $\approx 99.4\%$, IF: 100%), matching PEFTGUARD on IF and remaining essentially tied on SC. Classical backdoor detectors operate on *centralized* training signals (e.g., token statistics, attention cues, or model-internal traces) and assume direct access to *raw model updates/gradients*, making them *inapplicable* to federated PEFT of LLMs under secure aggregation. INDEXGUARD enables *FL-compatible* vetting—unsupervised, lightweight, and index-only—while matching the strongest centralized baseline in *average* detection; per-attack comparisons with *PEFTGuard* are deferred

*Table 3.* Detection performance compared to baselines. SC/IF averages are computed over non-toxic/ toxic datasets.

| Method | SC | | IF | |
|---|---|---|---|---|
| | Acc | AUC | Acc | AUC |
| T-Miner (USENIX'21) | 50% | 0.500 | – | – |
| AttenTD (NAACL'22) | 50% | 0.606 | – | – |
| PICCOLO (S&P'22) | 76% | 0.890 | – | – |
| MNTD (S&P'21) | 88% | 0.937 | 51% | 0.510 |
| PEFTGUARD | 99% | 1.000 | 100% | 1.000 |
| **INDEXGUARD (Ours)** | **99.39%** | **0.998** | **100%** | **1.000** |

*Table 4.* InsertSent detection across LLMs (AG News, SST-2)

| Model | AG News (Acc/ AUC) | SST-2 (Acc/ AUC) |
|---|---|---|
| Llama-2-7B | $100.00 \pm 0.00$ / $1.00 \pm 0.00$ | $99.83 \pm 0.24$ / $0.996 \pm 0.007$ |
| Llama-3.2-3B | $100.00 \pm 0.00$ / $1.00 \pm 0.00$ | $99.67 \pm 0.47$ / $1.000 \pm 0.00$ |
| Llama-3-8B | $99.67 \pm 0.47$ / $1.000 \pm 0.00$ | $100.00 \pm 0.00$ / $1.00 \pm 0.00$ |
| BERT-base | $99.33 \pm 0.47$ / $0.996 \pm 0.003$ | $99.00 \pm 0.58$ / $0.994 \pm 0.004$ |
| RoBERTa-base | $98.33 \pm 0.58$ / $0.992 \pm 0.004$ | $98.33 \pm 0.47$ / $0.991 \pm 0.005$ |

to the Appx.F. **Backbone- and adapter-agnostic detection.** Tables 4 and 5 evaluate whether INDEXGUARD's index-only vetting signal persists across (i) substantially different *base models* and (ii) different *LoRA-family parameterizations*, while fixing the trigger to InsertSent. Across base models (Table 4), decoder-only LLaMA variants remain at (or extremely close to) perfect separation on both AG News and SST-2 (AUC $\approx 1.0$ with accuracy $\geq 99.67\%$), indicating that the Top-$K$ salience index distribution is stable under changes in scale and architecture within the LLaMA family. Encoder-only backbones (BERT/RoBERTa) show a small but consistent drop (e.g., AUC $\approx 0.991$–0.996), suggesting reduced score margin under the same sketch budget, yet still yielding highly reliable vetting. Across PEFT methods on a fixed LLaMA-3 backbone (Table 5), detection is essentially unchanged from standard LoRA: all variants achieve near-ceiling AUC ($\geq$ 0.998) and accuracy ($\geq$ 99.33%), with several configurations recovering exact AUC= 1.0 on both tasks. Together, these sweeps show that INDEXGUARD is not model- or adapter-specific: robust malicious/benign separation is driven by index-only salience geometry and persists across diverse backbones/ PEFT parameterizations.

**Sensitivity to statistical heterogeneity (Dirichlet $\alpha$).** Table 6 shows that stronger label skew (smaller $\alpha$) modestly erodes index-only vetting on AG News. As $\alpha$ drops ($0.5 \rightarrow 0.1$), benign clients become more idiosyncratic, increasing natural client drift and tightening the separation margin between benign and poisoned salience-index supports—an established stressor for FL anomaly defenses where benign heterogeneity can mimic adversarial outliers. Accordingly, the loss is confined to the already non-ceiling attacks: RIPPLES falls from Acc 99.1% to 97.2% and Syntactic from 99.6% to 98.7%. High-signal triggers remain essentially perfectly separable, indicating INDEXGUARD is robust to moderate skew and degrades gracefully only under extreme heterogeneity and intrinsically subtle signatures.

**End-to-End Attack mitigation.** Figure 5 tracks CA and ASR over FL rounds for BERT-base/SST-2 and

*Table 5.* Effect of PEFT method on InsertSent attack detection.

| PEFT Method | AG News (Acc/ AUC) | | SST-2 (Acc/ AUC) | |
|---|---|---|---|---|
| LoRA | 100.00%±0.00% | 1.000±0.000 | 99.67%±0.047% | 0.998±0.003 |
| AdaLoRA | 100.00%±0.00% | 1.000±0.000 | 99.50%±0.50% | 0.998±0.002 |
| QLoRA | 99.67%±0.047% | 0.999±0.001 | 99.33%±0.031% | 0.998±0.002 |
| LoRA+ | 100.00%±0.00% | 1.000±0.000 | 99.83%±0.024% | 1.000±0.000 |
| DoRA | 100.00%±0.00% | 1.000±0.000 | 100.00%±0.00% | 1.000±0.000 |

*Table 6.* Effect of non-IID level (Dirichlet $\alpha$) on AG News

| Attack | Metric | Dir(0.1) | Dir(0.3) | Dir(0.5) |
|---|---|---|---|---|
| InsertSent | Acc | 99.67%±0.47% | 100.00%±0.00% | 100.00%±0.00% |
| | AUC | 0.997±0.003 | 1.000±0.000 | 1.000±0.000 |
| RIPPLES | Acc | 97.20%±0.54% | 98.20%±0.31% | 99.10%±0.33% |
| | AUC | 0.989±0.006 | 0.996±0.005 | 0.998±0.002 |
| Syntactic | Acc | 98.70%±0.33% | 99.30%±0.00% | 99.60%±0.24% |
| | AUC | 0.993±0.004 | 0.998±0.002 | 0.999±0.001 |
| StyleBkd | Acc | 99.33%±0.47% | 100.00%±0.00% | 100.00%±0.00% |
| | AUC | 0.995±0.004 | 1.000±0.000 | 1.000±0.000 |

Llama-2-7B/AG News. Without mitigation, ASR quickly saturates to 1.0 in both tasks, showing that low-budget poisoning compounds under repeated aggregation; CA may remain high but is compromised under triggered inputs. With mitigation, ASR stays low with mild fluctuations, while CA remains convergent with only a modest drop due to removing malicious clients and their data (0.921→0.895 on SST-2; 0.690→0.651 on AG News), yielding a favorable security–utility trade-off suitable for always-on deployment.

**Adaptive reported-support manipulation.** We also evaluate a dishonest-reporting adaptive attacker that trains a poisoned local model but reports a benign-looking Top-$K$ support to evade IOS-based clustering. Specifically, the attacker constructs the transmitted support by mixing clean and poisoned supports, where $b$ denotes the fraction of non-overlapping Top-$K$ indices drawn from the poisoned support. Smaller $b$ makes the transmitted support more benign-looking, while larger $b$ preserves more of the poisoned-support signal. Table 7 shows a clear evasion–attack-strength tradeoff. For small $b$ values, the attacker is harder to detect: INDEXGUARD obtains Acc./AUC of 71.2/0.55, 75.8/0.66, and 77.7/0.69 for $b = 0.1, 0.2, 0.3$, respectively. However, in these cases the unmitigated ASR@10 remains moderate, reaching only 55.1%, 58.3%, and 62.6%. As $b$ increases, the attack becomes stronger, with ASR@10 rising to 66.5% and 82.4% for $b = 0.4$ and $b = 0.5$, but detection also becomes much easier, reaching 96.7/0.94 and 100.0/1.00 Acc./AUC. Thus, before protected reporting is enforced, the attacker can either appear more benign with limited attack strength, or obtain a stronger backdoor at the cost of becoming highly detectable.

**Extended experiments.** Appx. F reports comparison to PEFTGUARD, poison-rate sensitivity, overhead breakdown, bandwidth and scalability analysis, and mixed-trigger-aware filtering, and adaptive reported-support manipulation.

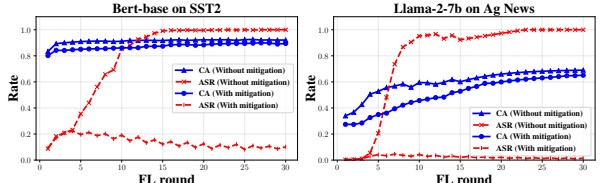

*Figure 5.* **Effect of attack mitigation on CA/ASR trajectories.** BERT-base on SST-2 and Llama-2-7B on AG News.

*Table 7.* Adaptive reported-support manipulation. The attacker trains a poisoned local model but reports a Top-$K$ support mixed with benign-looking indices. Smaller $b$ makes the reported support more benign-looking, while larger $b$ preserves more poisoned-support signal.

| $b$ ratio | INDEXGUARD Acc. (%) | INDEXGUARD AUC | ASR@10 (%, no mitigation) |
|---|---|---|---|
| 0.1 | 71.2 | 0.55 | 55.1 |
| 0.2 | 75.8 | 0.66 | 58.3 |
| 0.3 | 77.7 | 0.69 | 62.6 |
| 0.4 | 96.7 | 0.94 | 66.5 |
| 0.5 | 100.0 | 1.00 | 82.4 |

# 8. Limitations

We highlight two limitations. **(i)** While we evaluate INDEXGUARD across multiple PEFT variants, index-only vetting requires a *shared sketch space* across clients: a consistent monitored projection set and a common parameter-to-index mapping. This is natural in standard FL deployments where the PEFT configuration is fixed by design; however, if clients use incompatible adapter placements or substantially different parameterizations, additional alignment would be needed to make indices comparable. **(ii)** Our method is unsupervised and relies on relative cohesion/separation in IOS space; consequently, it is inherently less informative in a rare regime where the malicious population is the majority, where "minority vetting" needs side information.

# 9. Conclusion

We introduce INDEXGUARD, a lightweight index-only vetting layer in which clients transmit Top-$K$ salience index sketches over monitored PEFT projections; the server then builds the IOS overlap structure and uses unsupervised clustering to filter poisoned clients *before* aggregation. Across diverse backdoors, tasks, backbones, and PEFT variants—and under varying attacker prevalence and strong non-IID—INDEXGUARD delivers near-ceiling detection that translates into end-to-end mitigation, preserving clean accuracy while suppressing ASR. Overall, INDEXGUARD provides a practical, deployable primitive for secure federated customization of LLMs under realistic privacy constraints. Future work includes extending INDEXGUARD to more adaptive multi-trigger attackers and integrating protected reporting more tightly into the FL protocol. Combining index-only vetting with complementary defenses such as robust aggregation and deployment-time monitoring is another promising direction.

## Acknowledgment

This paper has received funding from the European Union's Horizon Europe research and innovation program under grant agreement No. 101178648. The European Commission's support for the production of this publication does not constitute an endorsement of the contents, which reflect the views only of the authors, and the Commission cannot be held responsible for any use which may be made of the information contained therein.

## Impact Statement

This work aims to improve the security and trustworthiness of federated parameter-efficient fine-tuning of large language models. By enabling index-only pre-aggregation vetting, INDEXGUARD can help detect and filter backdoored client updates without requiring the server to inspect per-client real-valued updates, which is important for privacy-sensitive deployments such as healthcare, enterprise, and cross-silo collaboration. The main positive impact is reducing the risk that malicious or compromised participants implant hidden trigger-controlled behavior while preserving clean utility and compatibility with secure aggregation. Potential negative impacts include false filtering of benign clients under extreme heterogeneity, over-reliance on a single defense, and possible attacker adaptation after detection signals are published. We therefore view INDEXGUARD as one component of a broader defense-in-depth pipeline, to be combined with secure aggregation, protected reporting, monitoring, and careful deployment-specific validation.

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

## A. Top-$K$ budget selection via coverage and stability

**PEFT-specific setup.** Let $\theta_{\mathcal{P}}$ denote the monitored trainable PEFT parameters (LoRA on $Q/V$ only), with $M := |\theta_{\mathcal{P}}|$ and fixed global coordinate IDs. At round $t$, client $i$ forms $\Delta\theta_i^{(t)} = \theta_{i,\mathcal{P}}^{(t+1)} - \theta_{\mathcal{P}}^{(t)}$ and sets salience $s_i^{(t)}[m] = |\Delta\theta_i^{(t)}[m]|$. We assign a fixed global index to each coordinate in $\theta_{\mathcal{P}}$ (shared across clients/rounds), which defines the index-only sketch space.

**Coverage and stability criteria.** Sort indices $j_1, \ldots, j_M$ by descending $s_{i,j}$. Define monotone cumulative coverage

$$C_i(K) = \frac{\sum_{t=1}^{K} s_{i,j_t}}{\sum_{t=1}^{M} s_{i,j_t}} \in [0,1], \tag{5}$$

and an overlap-stability score using $r$ lightweight resamples $\{s_i^{(b)}\}_{b=1}^r$ (mini-batch bootstrap or adjacent time windows):

$$\rho_i(K) = \frac{2}{r(r-1)} \sum_{b<b'} \frac{\left| \text{TopK}(s_i^{(b)}, K) \cap \text{TopK}(s_i^{(b')}, K) \right|}{K} \in [0,1]. \tag{6}$$

Let $K_i^{\text{cov}}$ be the smallest $K \leq K_{\max}$ such that $C_i(K) \geq \tau$. If stability is enforced, we set $K_i^\star = \min\{K \in [K_i^{\text{cov}}, K_{\max}] : \rho_i(K) \geq \rho_0\}$; if no such $K$ exists, we set $K_i^\star = K_{\max}$. Because $C_i(K)$ is monotone, we find $K_i^{\text{cov}}$ by binary search. Since $\rho_i(K)$ is not guaranteed to be monotone, we then scan upward from $K_i^{\text{cov}}$ until the stability constraint is met (or $K_{\max}$ is reached).

**Server-side global $K$ and optional block budgets.** To keep IOS normalization consistent, the server fixes a single $K$ for the round by aggregating $\{K_i^\star\}$ (median; clipped to $K_{\max}$) and broadcasts it. Clients then transmit $\text{TopKIdx}(s_i, K)$; if $K > K_i^\star$ they append the next-highest-salience indices, and if $K < K_i^\star$ they truncate. If needed, we can allocate $K$ per block (per layer or per $\{Q, V\}$ projection) with $\sum_\ell K_\ell = K$ to avoid cross-block domination.

**Complexity.** Binary search needs $O(\log K_{\max})$ iterations; each iteration uses selection/prefix sums once the ordering is available. We sort once to obtain the ordering/prefix sums, costing $O(M \log M)$ time; extracting TopKIdx thereafter is $O(K)$. Once a global $K$ is fixed (e.g., from warm-up), per-round extraction of $\text{TopKIdx}(s_i, K)$ need not sort all $M$ coordinates: it can be implemented via partial selection (Quickselect/`nth_element`), which finds the Top-$K$ set in $O(M)$ expected time (and $O(K \log K)$ if the Top-$K$ indices must be ordered).

**Additional experiments.** To justify that our chosen $K$ is not a tuned knob, we report: (i) **Sensitivity to $(\tau, \rho_0, r)$:** grid $\tau \in \{0.85, 0.9, 0.95\}$, $\rho_0 \in \{0.7, 0.8, 0.9\}$, $r \in \{3, 5\}$ and measure CA/ASR/AUROC and vetting overhead; (ii) **Fixed-$p$ baselines:** compare $K = K^\star$ to $K = pM$ with $p \in \{1\%, 2\%, 5\%, 10\%\}$; (iii) **When to compute $K$:** warm-up only (compute $K$ in the first round and keep fixed) vs. recompute every $R$ rounds. As shown in the IOS study, using the coverage+stability knee ($K^\star$) typically dominates larger fixed fractions, while remaining lightweight.

## B. Index-leakage

## C. Time and bandwidth overhead (detailed)

This appendix expands the overhead discussion by making explicit (i) the per-round compute components introduced by Phase A (index sketching) and (ii) the additional bandwidth and uplink latency relative to the secure-aggregation payload.

**Notation and per-round baselines.** Let $M$ denote the number of *monitored* PEFT coordinates (i.e., the flattened dimensionality of the Q/V adapter parameters in the canonical monitored space), and let $K$ be the sketch size (Top-$K$ indices) with $K \ll M$. Let $b_{\text{val}}$ be the number of bits used to transmit a single update value in the secure-aggregation payload (e.g., 16–32 bits depending on quantization/packing), and $b_{\text{idx}}$ the number of bits per index (typically $b_{\text{idx}} = \lceil \log_2 M \rceil$ plus a small header/packing overhead). We denote by $R$ the client uplink rate (bits/s). Per round $t$, the *baseline* FL time without INDEXGUARD is

$$T_{\text{base}}^{(t)} = T_{\text{train}}^{(t)} + T_{\text{model-comm}}^{(t)}, \tag{7}$$

where $T_{\text{train}}^{(t)}$ is local PEFT training and $T_{\text{model-comm}}^{(t)}$ is the time to transmit the adapter/model payload under secure aggregation.

**Phase A compute: score extraction and Top-$K$.** Phase A adds instrumentation to form an index-only sketch from the client update in round $t$: (1) compute an importance score per monitored coordinate (e.g., coordinate-wise magnitude of the PEFT update restricted to the monitored Q/V space), and (2) select the Top-$K$ indices. Both steps are linear in the monitored footprint $M$:

$$T_{\text{sig-calc}}^{(t)} = T_{\text{score}}^{(t)} + T_{\text{topK}}^{(t)} = O(M). \tag{8}$$

In practice, the score pass is a single streaming pass over the update tensor (read + abs/accumulate), and Top-$K$ can be implemented via a linear-time selection routine or a heap/partial-sort with near-linear behavior for small $K$. Moreover, this cost is *amortized* relative to local training because (i) it touches only the adapter parameters (not the full model) and (ii) it is executed once per round, whereas training performs multiple forward/backward passes. We therefore write the total added compute time as

$$T_{\text{A}}^{(t)} \equiv T_{\text{sig-calc}}^{(t)} = O(M) \quad \text{(amortized by PEFT training).} \tag{9}$$

**Phase A bandwidth: index-only uplink.** Phase A communicates only the Top-$K$ *index set* (no values), hence the per-round added uplink is

$$B_{\text{A}}^{(t)} = K\, b_{\text{idx}} \text{ bits.} \tag{10}$$

By contrast, the secure-aggregation payload for the PEFT update transmits values for the monitored footprint (or the adapter payload being aggregated), which scales as

$$B_{\text{agg}}^{(t)} \approx M\, b_{\text{val}} \text{ bits,} \tag{11}$$

up to protocol headers and packing. The relative bandwidth overhead is therefore

$$\frac{B_{\text{A}}^{(t)}}{B_{\text{agg}}^{(t)}} \approx \frac{K\, b_{\text{idx}}}{M\, b_{\text{val}}}, \tag{12}$$

which is negligible when $K \ll M$. The additional uplink latency introduced by Phase A is approximately

$$\Delta T_{\text{comm}}^{(t)} \approx \frac{B_{\text{A}}^{(t)}}{R} = \frac{K\, b_{\text{idx}}}{R}. \tag{13}$$

**End-to-end per-round overhead decomposition.** Combining compute and communication, the total additional round time introduced by INDEXGUARD can be decomposed as

$$T_{\text{oh}}^{(t)} = T_{\text{sig-calc}}^{(t)} + T_{\text{sig-comm}}^{(t)} + T_{\text{vet}}^{(t)}, \tag{14}$$

where $T_{\text{sig-comm}}^{(t)} \approx \Delta T_{\text{comm}}^{(t)}$ is the index-uplink time and $T_{\text{vet}}^{(t)}$ is the server-side overlap-based vetting time. The server-side vetting consists only of set intersections / overlap scoring over received Top-$K$ sets and therefore scales with the sketch size (and number of compared pairs), not with the model size; for fixed $K$ it is typically millisecond-level (cf. Table 13). The overall per-round wall-clock time with INDEXGUARD is then

$$T_{\text{tot}}^{(t)} = T_{\text{base}}^{(t)} + T_{\text{oh}}^{(t)} = \left(T_{\text{train}}^{(t)} + T_{\text{model-comm}}^{(t)}\right) + \left(T_{\text{sig-calc}}^{(t)} + T_{\text{sig-comm}}^{(t)} + T_{\text{vet}}^{(t)}\right). \tag{15}$$

**Why overhead does not grow with full-model size.** A key point is that Phase A operates on the *PEFT adapter footprint* (monitored Q/V coordinates) rather than full-model parameters: both $T_{\text{sig-calc}}^{(t)}$ and $B_{\text{A}}^{(t)}$ scale with $(M, K)$. Thus, even as the backbone scales from encoder-only models to multi-billion-parameter LLMs, overhead remains small provided the adapter footprint and sketch size stay moderate and $K \ll M$. This is corroborated empirically by the per-round measurements in Table 13 and the bandwidth sweep in Table 14.

**Practical parameterization.** In typical deployments, $b_{\text{idx}}$ is on the order of tens of bits (since $M$ is the monitored adapter dimensionality), while $b_{\text{val}}$ is 16–32 bits (or larger if uncompressed). Therefore, the ratio $\frac{K\, b_{\text{idx}}}{M\, b_{\text{val}}}$ is dominated by $K/M$; as long as $K/M$ is in the single-digit percent range, the index-only uplink constitutes a negligible fraction of the secure-aggregation payload and does not materially affect round time.

Empirically, Appendix F.4 confirms these asymptotic predictions: across backbones, the measured $T_{\text{sig-calc}}$, $T_{\text{sig-comm}}$, and millisecond-level $T_{\text{vet}}$ remain negligible relative to $T_{\text{train}}$ and $T_{\text{model-comm}}$, yielding near-zero impact on end-to-end round time.

# D. Extended Theoretical Analysis

We now provide the full statements and proof sketches for Sec. 6.

## D.1. IOS foundations: alignment, stability, and robustness

We summarize three properties of Top-$K$ overlap that justify IOS as a similarity primitive for index-only vetting.

**Setup.** Let $s, t \in \mathbb{R}_{\geq 0}^M$ be nonnegative salience vectors (absolute PEFT-update magnitudes), and define $S_s := \text{TopK}(s)$, $S_t := \text{TopK}(t)$ and $IOS(s, t) := |S_s \cap S_t|/K$. Since salience is nonnegative, IOS captures agreement in *which coordinates are salient* (not update sign).

### D.1.1. ALIGNMENT: OVERLAP LOWER-BOUNDS SHARED SALIENCE MASS

We assume Top-$K$ entries are not arbitrarily dispersed.

**Assumption D.1** (Mass concentration). $\sum_{j \in S_s} s_j \geq \tau_s \|s\|_1$ and $\sum_{j \in S_t} t_j \geq \tau_t \|t\|_1$ for some $\tau_s, \tau_t \in (0, 1]$.

**Assumption D.2** (Bounded dispersion within Top-$K$). For all $j \in S_s$, $s_j \in [\underline{s}, \overline{s}]$ and for all $j \in S_t$, $t_j \in [\underline{t}, \overline{t}]$ with $\kappa_s := \overline{s}/\underline{s}$ and $\kappa_t := \overline{t}/\underline{t}$ bounded.

**Proposition D.3** (Support overlap implies shared salience mass (conditional)). *Let $s, t \in \mathbb{R}_{\geq 0}^M$ with supports $S_s, S_t$. If $IOS(s, t) \geq \rho$, then under Assumption D.2,*

$$\sum_{j \in S_s \cap S_t} s_j \geq \frac{\rho}{\kappa_s} \sum_{j \in S_s} s_j, \qquad \sum_{j \in S_s \cap S_t} t_j \geq \frac{\rho}{\kappa_t} \sum_{j \in S_t} t_j. \tag{16}$$

*With Assumption D.1, the overlap carries at least $\frac{\rho \tau_s}{\kappa_s} \|s\|_1$ and $\frac{\rho \tau_t}{\kappa_t} \|t\|_1$ salience mass, respectively.*

*Proof sketch.* Since $|S_s \cap S_t| \geq \rho K$ and $s_j \geq \underline{s}$ for $j \in S_s$ (Assumption D.2),

$$\sum_{j \in S_s \cap S_t} s_j \geq |S_s \cap S_t| \underline{s} \geq \rho K \underline{s}.$$

Also $\sum_{j \in S_s} s_j \leq K\overline{s} = K\kappa_s \underline{s}$, yielding $\sum_{j \in S_s \cap S_t} s_j \geq (\rho/\kappa_s) \sum_{j \in S_s} s_j$. The bound for $t$ is identical. Combining with Assumption D.1 gives the $\ell_1$ statements. $\square$

### D.1.2. STABILITY: TOP-$K$ SUPPORTS ARE REPRODUCIBLE UNDER MILD NOISE

Let $s_i^{(t)}$ be the salience at round $t$ and assume

$$s_i^{(t)} = \mu_i + \xi_i^{(t)}, \tag{17}$$

where $\mu_i$ is a stationary mean salience vector and $\xi_i^{(t)}$ has independent, mean-zero sub-Gaussian coordinates with proxy variance $\sigma_i^2$. Define the Top-$K$ boundary margin $\Delta_i := \mu_{i,(K)} - \mu_{i,(K+1)} > 0$ (ties broken deterministically).

Optionally, let $\tilde{s}_i^{(t)}$ be an EMA-smoothed estimator with $\beta \in [0, 1)$:

$$\tilde{s}_i^{(t)} = \beta \tilde{s}_i^{(t-1)} + (1 - \beta) s_i^{(t)}. \tag{18}$$

EMA reduces the effective noise scale; below we present a conservative bound based on a union bound over coordinates.

**Theorem D.4** (Top-$K$ stability under bounded noise (sufficient condition)). *Let $s \in \mathbb{R}_{\geq 0}^M$ with sorted entries $s_{(1)} \geq \cdots \geq s_{(M)}$ and Top-$K$ margin $\Delta := s_{(K)} - s_{(K+1)}$. Let $\hat{s} = s + \varepsilon$ where $\varepsilon_j$ are independent sub-Gaussian coordinates of scale $\sigma$. If $\max_j |\varepsilon_j| \leq \Delta/2$, then $\text{TopK}(\hat{s}) = \text{TopK}(s)$. Moreover,*

$$\Pr[\text{TopK}(\hat{s}) \neq \text{TopK}(s)] \leq 2M \exp\left(-\frac{\Delta^2}{8\sigma^2}\right). \tag{19}$$

*Proof sketch.* If $|\varepsilon_j| \leq \Delta/2$ for all $j$, then every true Top-$K$ entry remains above every true non-Top-$K$ entry after perturbation. For the probability bound, use the sub-Gaussian tail $\Pr(|\varepsilon_j| > \Delta/2) \leq 2 \exp(-\Delta^2/(8\sigma^2))$ and union bound over $M$ coordinates. $\square$

D.1.3. ROBUSTNESS: INVARIANCE TO SCALING AND RESILIENCE TO DIAGONAL PRECONDITIONING

Since $S_i$ depends only on rank order, IOS is invariant to global scaling and resilient to bounded diagonal rescaling.

**Lemma D.5** (Effect of bounded diagonal rescaling on Top-$K$). *Let $s \in \mathbb{R}_{\geq 0}^M$ and $D = (d_1, \ldots, d_M)$ with $d_j \in [1-\delta, 1+\delta]$ for all $j$ and $0 \leq \delta < 1$. Let $\tilde{s} = Ds$ and $\Delta := s_{(K)} - s_{(K+1)}$. If $\Delta > 2\delta s_{(K)}$, then $\mathrm{TopK}(\tilde{s}) = \mathrm{TopK}(s)$. More generally, membership changes can only occur among coordinates in an $O(\delta)$ band around the threshold: $s_j \in [s_{(K)}/(1+\delta), s_{(K)}/(1-\delta)]$.*

*Proof sketch.* A Top-$K$ element can decrease by at most $(1-\delta)$ while a non-Top-$K$ element can increase by at most $(1+\delta)$. Thus Top-$K$ is preserved if $(1-\delta)s_{(K)} > (1+\delta)s_{(K+1)}$, implied by $\Delta > 2\delta s_{(K)}$. The band statement follows by solving the flip inequalities. $\square$

## D.2. Separation for backdoor detection under a shared-trigger imprint

We connect IOS to backdoor-client isolation. Let $\mathcal{B}$ denote benign clients (task objective) and $\mathcal{A}$ adversarial clients (task+backdoor) that share a trigger. Define group-wise expected IOS: $IOS_{\mathcal{BB}} := \mathbb{E}[IOS_{ij} \mid i, j \in \mathcal{B}]$, $IOS_{\mathcal{AA}} := \mathbb{E}[IOS_{ij} \mid i, j \in \mathcal{A}]$, $IOS_{\mathcal{BA}} := \mathbb{E}[IOS_{ij} \mid i \in \mathcal{B}, j \in \mathcal{A}]$.

**Randomness and observable supports.** Fix round $t$ and let each client report $S_i := \mathrm{TopK}(s_i^{(t)})$ where $s_i^{(t)} = \mu_i + \xi_i^{(t)}$. Let $S_i^\star := \mathrm{TopK}(\mu_i)$ be the noise-free Top-$K$ support and define the stability event $E_i := \{S_i = S_i^\star\}$. For benign clients define $\delta_{\mathcal{B}} := \sup_{i \in \mathcal{B}} \Pr[E_i^c]$; analogously, $\delta_{\mathcal{A}}$ will capture adversarial instability with respect to the imprint set defined below. By Theorem D.4, $\delta$ is small when the Top-$K$ margin dominates the (possibly EMA-reduced) noise scale.

**Benign heterogeneity under non-IID data.** Under non-IID data, benign supports differ across clients. Let

$$\rho_{\mathcal{B}} := \mathbb{E}\left[\frac{|S_i^\star \cap S_j^\star|}{K} \,\middle|\, i \neq j, \, i, j \in \mathcal{B}\right]. \tag{20}$$

**Shared-trigger imprint.** A shared trigger induces a shared auxiliary objective among adversaries, amplifying a *common imprint set* $T \subseteq [M]$ of salient coordinates. This yields higher adversary–adversary Top-$K$ overlap even though both groups train on text.

**Assumption D.6** (Trigger imprint and benign exclusivity). There exist $T \subseteq [M]$ with $|T| = K_T \leq K$, and parameters $\eta \in [0, 1)$ and $\delta_{\mathcal{A}} \in [0, 1)$ such that: (i) (**Adversarial imprint**) for any $i \in \mathcal{A}$, $\Pr[T \subseteq S_i] \geq 1 - \delta_{\mathcal{A}}$; (ii) (**Benign exclusivity**) for any $i \in \mathcal{B}$, $\mathbb{E}[|T \cap S_i|]/K \leq \eta$.

Assumption D.6 is implied when trigger-imprint coordinates have a positive mean margin above the Top-$K$ boundary for adversaries, in which case Theorem D.4 yields small $\delta_{\mathcal{A}}$.

**Main result: an IOS cohesion gap.** Let

$$\rho_{\mathcal{BA}} := \mathbb{E}\left[\frac{|S_i^\star \cap S_j^\star|}{K} \,\middle|\, i \in \mathcal{B}, \, j \in \mathcal{A}\right]$$

denote the noise-free task-driven benign–adversary overlap.

**Theorem D.7** (IOS gap: high adversarial cohesion and controlled cross-group similarity). *Assume $\Pr[S_i \neq S_i^\star] \leq \delta_{\mathcal{B}}$ for $i \in \mathcal{B}$ (e.g., via Theorem D.4) and Assumption D.6 for $i \in \mathcal{A}$. Then:*

$$IOS_{\mathcal{AA}} \geq (1 - 2\delta_{\mathcal{A}})\frac{K_T}{K}, \tag{21}$$

$$IOS_{\mathcal{BA}} \leq \rho_{\mathcal{BA}} + \eta + \delta_{\mathcal{A}}, \tag{22}$$

$$IOS_{\mathcal{BB}} \leq \rho_{\mathcal{B}} + 2\delta_{\mathcal{B}}. \tag{23}$$

*In particular, if $(1 - 2\delta_{\mathcal{A}})\frac{K_T}{K} > \rho_{\mathcal{B}} + 2\delta_{\mathcal{B}}$, then $IOS_{\mathcal{AA}} > IOS_{\mathcal{BB}}$ (strict cohesion gap), while benign–adversary similarity is controlled by (22).*

*Proof sketch.* For $i, j \in \mathcal{A}$, with probability at least $1 - 2\delta_{\mathcal{A}}$ both include $T$, hence $|S_i \cap S_j| \geq |T| = K_T$, giving (21). For $i \in \mathcal{B}, j \in \mathcal{A}$, decompose $|S_i \cap S_j|$ into task-driven overlap plus imprint overlap; taking expectations and using Assumption D.6(ii) yields (22). For $i, j \in \mathcal{B}$, stability implies observed supports match $S^{\star}$ except with probability $\delta_{\mathcal{B}}$, giving (23). □

**Operational meaning and design implications.** $\delta_{\mathcal{A}}, \delta_{\mathcal{B}}$ quantify Top-$K$ instability and decrease when the Top-$K$ margin $\Delta$ dominates the effective noise scale $\sigma$ (Theorem D.4); monitoring PEFT projections where salience concentrates increases margins and improves stability. The separation condition is controlled by $K_T/K$ and $\delta_{\mathcal{A}}, \delta_{\mathcal{B}}$, while benign heterogeneity $\rho_{\mathcal{B}}$ sets the baseline overlap under non-IID data. These parameters motivate selecting (i) monitored projections $\mathcal{P}$ with concentrated salience and (ii) $K$ trading off imprint capture ($K_T/K$) against benign overlap ($\rho_{\mathcal{B}}$).

**Scope and adaptive attackers.** Theorem D.7 formalizes the shared-trigger regime. Fully adaptive adversaries that intentionally de-cohere supports (e.g., diversified supports, padding with benign-like coordinates, multiple triggers/targets) can reduce cohesion gaps and weaken clustering-based vetting; this is complementary to our focus and left to future work.

# E. Privacy Enhancement

**Robust solution against leakage from $S_i$.** We instantiate the round-specific salt using a lightweight client-side derivation under a shared group key. During client enrollment (or a one-time setup phase), participating clients obtain a symmetric group key $\mathbb{K}$ (e.g., provisioned by the FL coordinator or established via a standard distributed key agreement (Kate & Goldberg, 2009)). At round $t$, each client derives a salt from pseudorandom function $\mathrm{PRF}$ as

$$\mathsf{salt}_t := \mathrm{PRF}(\mathbb{K}, t), \tag{24}$$

where $\mathrm{PRF}$ can be implemented as HMAC (e.g., HMAC-SHA256). The client then replaces each raw index $S_{i,j}^{(t)} \in S_i^{(t)}$ with a commitment token using $Hash$ where $H : \{0,1\}^* \to \{0,1\}^\kappa$ is a cryptographic hash function as:

$$h_{i,j}^{(t)} := Hash(S_{i,j}^{(t)} \parallel \mathsf{salt}_t), \tag{25}$$

and sends $\tilde{S}_i^{(t)} := \{h_{i,j}^{(t)}\}$ to the server instead of $S_i^{(t)}$. The server computes IOS and clustering using the tokens exactly as in the raw-index pipeline. $\tilde{S}_i^{(t)}$ does not reveal any information about $S_i^{(t)}$ and prevent the corrupted server from mapping tokens to specific layers/parameters. However, note that if the server and a set of clients collude, other benign clients can be vulnerable since the shared $\mathbb{K}$ as well as local states (e.g., $\tilde{S}_i^{(t)}, S_i^{(t)}, \theta_i, \mathcal{D}_i$) of the colluding clients will be revealed to the server. We plan to investigate protection against such collusion in the future.

**Accuracy and overhead.** Salting does not change IOS values within a round because intersection sizes are preserved under a common $\mathsf{salt}_t$, hence detection/mitigation accuracy remains unchanged up to negligible hash collisions. Overhead is lightweight: each client performs hash operations per round (for $|S_i^{(t)}|$) plus a single PRF evaluation, and communication remains $O(\mathbb{K})$ tokens per round (same order as raw indices, with token size fixed by the hash output).

**Security of INDEXGUARD against malicious clients.** While our approach protects each client report $S_i$, it weakens the server ability to enforce correctness of the client side computation. This introduces two integrity gaps. First, a malicious client may submit an arbitrary report that is not produced by $\theta_i$. In our setting, this corresponds to sending digests $\{h_{i,j}\}_j$ that do not bind to any vector $S_i$ satisfying $S_i = F(\theta_i, \xi_i)$. Second, even if the client submits a report that is consistent with some local computation, it may use a local model $\theta_i$ trained on poisoned data $\tilde{\mathcal{D}}_i$, which undermines the intended semantics of honest local training.

We address the first gap using verifiable computation, and specifically zero knowledge proofs (ZKPs) (Goldwasser et al., 2019). Informally, a ZKP enables a prover to convince a verifier that a public statement is true with respect to some secret witness, without revealing the witness. Formally, let $\mathcal{R}$ be an NP relation. The prover holds a witness $w$ and the verifier holds a statement $x$. The prover convinces the verifier that $\mathcal{R}(x, w) = 1$ while revealing no additional information about $w$ beyond the validity of the claim.

In our setting, the client plays the role of the prover and the server plays the role of the verifier. The public statement contains the round context $\xi_i$, a commitment to the local model, and per component salted hash digests of the client report,

namely $h_{i,j} = H(\mathsf{salt}_{i,j} \| S_{i,j})$ for all $j$. The witness contains $\theta_i$, the report vector $S_i = (S_{i,0}, \dots, S_{i,\ell-1})$, and the salts $\{\mathsf{salt}_{i,j}\}_j$. The proof convinces the server that the client knows such values and that they satisfy the prescribed computation $S_i = F(\theta_i, \xi_i)$ and the digest consistency constraints for every component, without revealing $\theta_i$, $S_i$, or the salts.

We instantiate this mechanism using zkSNARKs, which are succinct non interactive arguments of knowledge with a zero knowledge property (Ben-Sasson et al., 2014; Liang et al., 2025). zkSNARKs produce short proofs and enable fast verification, which is critical for scalability in federated learning. At a high level, we encode the intended client computation as an arithmetic circuit that checks the model commitment opening, the equality $S_i = F(\theta_i, \xi_i)$, and the per component digest equations $h_{i,j} = H(\mathsf{salt}_{i,j} \| S_{i,j})$.

Finally, we note that zkSNARKs enforce correctness with respect to the specified relation. They do not by themselves guarantee that $\theta_i$ was trained on clean data.

We now delve into integrating zero knowledge proof to INDEXGUARD. Let $\Theta$ be the space of local models and let the client report space be $\mathcal{S} = \mathcal{S}_0^\ell$ for some length $\ell = |S_i|$. Let $F : \Theta \times \mathcal{X} \to \mathcal{S}$ be a deterministic algorithm that outputs

$$S_i := F(\theta_i, \xi_i) = (S_{i,0}, \dots, S_{i,\ell-1}),$$

from a local model $\theta_i \in \Theta$ and public context $\xi_i \in \mathcal{X}$. Let $\mathsf{Com}$ be a computationally hiding and binding commitment scheme (used for $\theta_i$).

Client $i$ samples independent secret salts $\{\mathsf{salt}_{i,j}\}_{j=0}^{\ell-1}$ and publishes the per-component salted digests

$$h_{i,j} := Hash(\mathsf{salt}_{i,j} \| S_{i,j}) \quad \text{for } j \in \{0, \dots, \ell-1\},$$

collecting them as $\mathbf{h}_i := (h_{i,0}, \dots, h_{i,\ell-1})$, which are sent to the server as the verifier. We now define the NP relation $\mathcal{R}_F^{\mathbf{H}}$ over public statements

$$x := (i, \xi_i, C_{M,i}, \mathbf{h}_i)$$

and witnesses

$$w := \big(\theta_i, r_{M,i}, (S_{i,0}, \dots, S_{i,\ell-1}), (\mathsf{salt}_{i,0}, \dots, \mathsf{salt}_{i,\ell-1})\big)$$

by

$$\mathcal{R}_F^{\mathbf{H}}(x, w) = 1 \iff \Big(C_{M,i} = \mathsf{Com}(\theta_i; r_{M,i}) \wedge (S_{i,0}, \dots, S_{i,\ell-1}) = F(\theta_i, \xi_i) \wedge \bigwedge_{j=0}^{\ell-1} h_{i,j} = Hash(\mathsf{salt}_{i,j} \| S_{i,j})\Big).$$

Let $C_{F,\mathbf{H}}$ be an arithmetic circuit (e.g., R1CS) that decides $\mathcal{R}_F^{\mathbf{H}}$.

**zkSNARK system.** A *zkSNARK-based per-component consistency scheme* is a tuple of probabilistic polynomial time algorithms $\Pi_{F,\mathbf{H}} = (\mathsf{Setup}, \mathsf{Prove}, \mathsf{Verify})$ such that

$$(\mathsf{pk}, \mathsf{vk}) \leftarrow \mathsf{Setup}(1^\lambda, C_{F,\mathbf{H}}), \qquad \pi_i \leftarrow \mathsf{Prove}(\mathsf{pk}, x, w), \qquad 0/1 \leftarrow \mathsf{Verify}(\mathsf{vk}, x, \pi_i).$$

**Security (informal).** $\Pi_{F,\mathbf{H}}$ satisfies completeness, knowledge soundness (with an extractor outputting a witness $w$ for any client as a prover that produces an accepting proof), and zero knowledge (there exists a simulator outputting proofs indistinguishable from real ones given only $(\mathsf{vk}, x)$).

**Privacy interpretation.** The server observes only $(C_{M,i}, \mathbf{h}_i)$ and $\xi_i$; by zero knowledge and the hiding of $\mathsf{Com}$, it learns neither $\theta_i$ nor the vector $S_i$ nor the salts, while being assured that the published digests $\mathbf{h}_i$ bind to a report $S_i$ that is consistent with some committed model $\theta_i$ under $F$. After verifying these statements, the server can proceed to the private aggregation.

**Performance Evaluation.** Our zero-knowledge component is implemented as a Groth16 zk-SNARK (Groth, 2016). The arithmetic circuit is written in Circom, while setup, proof generation, and verification are handled using `snarkjs`. The purpose of this component is to verify the correctness of the reported Top-(K) support without revealing the underlying client tensor. Concretely, for client (i), the private witness consists of the hidden tensor $(u_i)$, whereas the public statement consists

of the reported Top-(K) index set $(S_i)$ and a public commitment to $(u_i)$. In the current prototype, the indices are public, reflecting the setting in which the server is allowed to observe the reported Top-(K) coordinates, while the corresponding tensor values remain hidden.

The implementation follows the standard offline/online separation of Groth16. In the offline phase, the circuit is compiled into an R1CS constraint system together with a witness generator. A circuit-specific proving key and verification key are then derived from a reusable Powers of Tau file. This setup is a one-time cost for a fixed circuit parameterization and can be reused across clients and FL rounds; it is not repeated in every round. In the online phase, each client locally computes its Top-(K) indices, generates a witness for its private tensor, proves that the published set $(S_i)$ is correct with respect to the hidden tensor, and sends the proof to the server. The server verifies the proof using only the public indices $(S_i)$, the public commitment, and the verification key, without learning the plaintext tensor values.

In our prototype, the circuit proves that the disclosed indices correspond exactly to the Top-(K) indices of the hidden tensor under a deterministic ranking rule. The compiled circuit contains 6,318 non-linear constraints and 715 linear constraints, with 32 private inputs, 4 public inputs, and 1 public output. In local experiments, proof generation takes approximately 4.02 seconds per client, while verification takes approximately 0.86 seconds. The offline setup phase takes approximately 7.1 seconds, assuming an existing Powers of Tau file. As expected for Groth16, proof generation constitutes the dominant online cost and is incurred by the client, whereas verification is substantially cheaper and therefore suitable for server-side checking. Overall, these results show that verifiable Top-(K) reporting is feasible at prototype scale and can serve as a concrete secure extension of the core pipeline.

## F. Additional Experimental Details

### F.1. Experimental Setup

**Backbones and PEFT.**    We evaluate both encoder-only backbones (**BERT-base** (Devlin et al., 2019), **RoBERTa-base** (Liu et al., 2019)) and decoder-only LLMs (**Llama-2-7B** (Touvron et al., 2023), **Llama-3-3B** (Llama Team, 2024), **Llama-3-8B** (Llama Team, 2024)) under LoRA-family PEFT: **LoRA** (Hu et al., 2022), **QLoRA** (Dettmers et al., 2023), **LoRA+** (Hayou et al., 2024), **AdaLoRA** (Zhang et al., 2023), and **DoRA** (Liu et al., 2024). Unless stated otherwise, all backbones are frozen and only PEFT parameters are trained. Adapters are injected into self-attention projections (`q_proj`, `v_proj` for decoder-only; `query`, `value` for encoder-only). For all PEFT variants, INDEXGUARD forms Top-$K$ index sketches in the *canonical monitored* Q/V coordinate space so indices are comparable across clients and rounds.

**Tasks, datasets, and formatting.**    We use two standard text-classification benchmarks, **SST-2** (Socher et al., 2013) and **AG News** (Zhang et al., 2015), and two instruction-following toxic backdoor corpora, `toxic-backdoors-alpaca` and `toxic-backdoors-hard` (derived from Alpaca-style instruction data (Taori et al., 2023)) (Ewart, 2024a;b). For encoder-only backbones (BERT/RoBERTa), SST-2/AG News are trained and evaluated with standard discriminative classification. For decoder-only LLMs (Llama-2/Llama-3), we cast SST-2/AG News as *generation-based classification* using a verbalizer prompt where the model outputs the label string.

**Backdoor injection and attack methods.**    Malicious clients generate backdoored adapters by poisoning a fraction $\rho$ of their local training samples (default $\rho=5\%$ unless stated otherwise). We instantiate dataset-appropriate backdoors as follows:

- **IF datasets (`toxic-backdoors-*`).** `toxic-backdoors-alpaca` uses a special-token/rare-token trigger (e.g., "SolidGoldMagikarp") (Ewart, 2024a), while `toxic-backdoors-hard` uses a sentence-level trigger containing "Manchester United" (Ewart, 2024b). The attacker objective is to elicit toxic generations conditioned on the trigger.

- **SST-2 and AG News (targeted misclassification).** We use four standard textual backdoors: (i) **RIPPLES**-style rare-word triggers (Kurita et al., 2020) by inserting one word from $\{\texttt{cf},\texttt{tq},\texttt{mn},\texttt{bb},\texttt{mb}\}$; (ii) **InsertSent** (Dai et al., 2019) by inserting a fixed sentence trigger (e.g., "I watched this 3D movie with my friends last Friday."); (iii) **Syntactic triggers** (Qi et al., 2021a) produced via SCPN (Iyyer et al., 2018), using the template $S(SBAR)(,)(NP)(VP)(.)$; (iv) **StyleBkd** (Qi et al., 2021c) using a style-transfer model to rewrite inputs into the *biblical* style. The target label is "positive" for SST-2 and "World" for AG News.

**Non-IID partitions.**    Client data heterogeneity is modeled via Dirichlet label partitions $\text{Dir}(\alpha)$; we report $\alpha=0.3$ by default and sweep $\alpha$ in ablations where stated.

*Table 8.* Default PEFT hyperparameters used to produce client adapters.

| Backbone | Task family | $r$ | $\alpha$ | drop | lr | epochs | micro-bsz | grad-acc | maxlen |
|---|---|---|---|---|---|---|---|---|---|
| BERT-base (Devlin et al., 2019) | SST-2/AGNews | 8 | 16 | 0.05 | $1\times10^{-4}$ | 3 | 32 | 1 | 256 |
| RoBERTa-base (Liu et al., 2019) | SST-2/AGNews | 8 | 16 | 0.05 | $1\times10^{-4}$ | 3 | 32 | 1 | 256 |
| Llama-2-7B (Touvron et al., 2023) | IF + Gen-cls | 8 | 128 | 0.05 | $2\times10^{-4}$ | 3 | 4 | 16 | 512 |
| Llama-3-3B (Llama Team, 2024) | IF + Gen-cls | 8 | 128 | 0.05 | $2\times10^{-4}$ | 3 | 4 | 16 | 512 |
| Llama-3-8B (Llama Team, 2024) | IF + Gen-cls | 8 | 128 | 0.05 | $2\times10^{-4}$ | 3 | 4 | 16 | 512 |

Llama-3-3B uses the same PEFT hyperparameters as Llama-2-7B unless stated otherwise.

*Table 9.* Attack detection comparison ($N = 20$, $|\mathcal{A}| = 6$) with PEFTGUARD. TB-Alp/Hard: toxic-backdoors-alpaca/hard. Attacks: Word, Sentence, InsertSent, RIPPLES, Syntactic, StyleBkd.

| Dataset | Attack | OURS | | PEFTGUARD | |
|---|---|---|---|---|---|
| | | Acc | AUC | Acc | AUC |
| TB-Alp | Word | 100.00%±0.00% | 1.000±0.000 | 100.00%±0.00% | 1.000±0.000 |
| TB-Hard | Sentence | 100.00%±0.00% | 1.000±0.000 | 100.00%±0.00% | 1.000±0.000 |
| AG News | InsertSent | 100.00%±0.00% | 1.000±0.000 | 98.33%±0.47% | 0.996±0.004 |
| | RIPPLES | 98.20%±0.31% | 0.996±0.005 | 100.00%±0.00% | 1.000±0.000 |
| | Syntactic | 99.30%±0.00% | 0.998±0.002 | 99.33%±0.47% | 0.998±0.003 |
| | StyleBkd | 100.00%±0.00% | 1.000±0.000 | 100.00%±0.00% | 1.000±0.000 |
| IMDB | InsertSent | 100.00%±0.00% | 1.000±0.000 | 99.33%±0.47% | 0.997±0.004 |
| | RIPPLES | 99.40%±0.47% | 0.997±0.006 | 100.00%±0.00% | 1.000±0.000 |
| | Syntactic | 98.90%±0.00% | 1.000±0.000 | 99.00%±0.00% | 0.984±0.001 |
| | StyleBkd | 98.70%±0.54% | 0.989±0.009 | 99.67%±0.47% | 1.000±0.000 |

**Optimization hyperparameters.** Table 8 reports default per-client adapter-training hyperparameters. We use AdamW with linear warmup (warmup ratio 0.03) and linear decay, weight decay 0.01, and gradient clipping at 1.0 (default $\beta_1{=}0.9$, $\beta_2{=}0.99$). For QLoRA, we use 4-bit NF4 quantization with double-quantization and bfloat16 compute (Dettmers et al., 2023).Unless stated otherwise, INDEXGUARD uses index-only Top-$K$ salience signatures computed from adapter update magnitudes $|\Delta\theta|$ (restricted to monitored modules, e.g., LoRA $W_q/W_v$). We perform agglomerative clustering on the IOS-induced distance $d(i,j) = 1 - \text{IOS}(i,j)$ and compute each cluster cohesion as the average pairwise IOS within the cluster. Unless otherwise stated, we set the number of clusters to $L = 2$ for shared-trigger experiments and use the mixed-trigger-aware rule with larger $L$ when multiple attack families are present. Let $C_{\ell^\star}$ denote the majority cluster. A non-majority cluster $C_\ell$ is flagged as suspicious if it is a cohesion outlier and not too large. We fix the size cap $\beta = 0.7$ and the cohesion margin $\gamma = 0.05$ for all experiments.

### F.2. Point-by-point comparison to PEFTGUARD

Table 9 aligns our per-task/per-attack results with PEFTGUARD, a *centralized* attack detector trained on PADBench via supervised clustering. Despite using *no supervision* and exposing only *index-only* Top-$K$ salience sketches (no raw PEFT updates/values), INDEXGUARD matches PEFTGUARD at ceiling on the IF backdoor corpora (TB-Alp/Hard: Acc$=$ 100%, AUC$=$ 1.0) and remains competitive on supervised classification tasks. On AG News, INDEXGUARD achieves perfect detection on three attacks and near-perfect on RIPPLES (Acc $\approx$ 98.2%, AUC $\approx$ 0.996), while PEFTGUARD is slightly better on RIPPLES but slightly worse on InsertSent; both methods are essentially tied on Syntactic and StyleBkd. On IMDB, both remain near ceiling across attacks with small gaps in opposite directions depending on trigger family. Overall, this head-to-head shows that *supervised, value-heavy centralized detectors are not necessary* to reach state-of-the-art detection: index-only, unsupervised vetting can deliver comparable accuracy while remaining compatible with secure aggregation and decentralized PEFT where update values cannot be inspected.

**Effect of per-client poison rate.** Table 10 sweeps the per-malicious-client poison rate $\rho$ on AG News ($N{=}40$, fixed malicious fraction) to test INDEXGUARD under increasingly strong local backdoor training. Detection improves monotonically with $\rho$ for all attacks: both accuracy and AUC rise from already-high values at the baseline $\rho{=}5\%$ to (near-)perfect separation by $\rho \in [15\%, 20\%]$. This trend is expected because higher $\rho$ amplifies the backdoor objective's contribution to each poisoned client's update, increasing the concentration of salient indices in the trigger-correlated subspace and reducing overlap with benign salience patterns; the resulting index sketches form a tighter malicious cluster with larger margin. By $\rho{=}20\%$, three attacks reach exact $100/1.00$ and the remaining one (RIPPLES) is effectively saturated (A$\approx$ 99.8%, AUC$\approx$ 0.999), indicating that INDEXGUARD remains effective even at low-budget poisoning and rapidly strengthens as the attacker increases poisoning intensity.

**Effect of number of malicious clients.** Table 11 varies the malicious fraction on AG News ($N{=}40$, $f \in 5, 10, 15, 20\%$)

*Table 10.* Effect of per-malicious-client poison rate $\rho$ for AG News.

| Attack | $\rho=5\%$ Acc | AUC | $\rho=10\%$ Acc | AUC | $\rho=15\%$ Acc | AUC | $\rho=20\%$ Acc | AUC |
|---|---|---|---|---|---|---|---|---|
| InsertSent | 99.3 | 0.985 | 99.7 | 0.993 | 100.0 | 1.000 | 100.0 | 1.000 |
| RIPPLES | 98.1 | 0.992 | 99.0 | 0.996 | 99.6 | 0.998 | 99.8 | 0.999 |
| Syntactic | 99.0 | 0.994 | 99.5 | 0.997 | 99.8 | 0.999 | 100.0 | 1.000 |
| StyleBkd | 99.6 | 0.997 | 99.8 | 0.999 | 100.0 | 1.000 | 100.0 | 1.000 |

*Table 11.* Effect of malicious-client fraction $f$.

| Attack | $f=5\%$ Acc | AUC | $f=10\%$ Acc | AUC | $f=15\%$ Acc | AUC | $f=20\%$ Acc | AUC |
|---|---|---|---|---|---|---|---|---|
| InsertSent | 99.3 | 0.98 | 99.8 | 0.99 | 100.0 | 1.00 | 100.0 | 1.00 |
| RIPPLES | 97.6 | 0.993 | 97.9 | 0.994 | 98.1 | 0.995 | 98.2 | 0.996 |
| Syntactic | 98.9 | 0.996 | 99.1 | 0.997 | 99.2 | 0.998 | 99.3 | 0.998 |
| StyleBkd | 100.0 | 1.00 | 100.0 | 1.00 | 100.0 | 1.00 | 100.0 | 1.00 |

and shows INDEXGUARD is largely insensitive to attacker prevalence: AUC remains high across attacks (typically $\geq 0.99$) with only minor accuracy fluctuations. Small improvements at larger $f$ (most evident for InsertSent, and slightly for RIPPLES/Syntactic) are expected since more attackers form a denser, more self-consistent signature cluster in index space, sharpening separation from heterogeneous benign updates; at very small $f$ (e.g., 5%), the boundary is more sample-limited, yielding slightly lower yet still strong AUC. High-signal triggers (StyleBkd, InsertSent for $f \geq 10\%$) stay near-perfect, while subtler ones degrade gracefully, confirming robust vetting even at low attacker prevalence.

### F.3. Sensitivity to the size-cap $\beta$

We ablate the size-cap $\beta$ (used to avoid over-filtering when a cohesive outlier cluster is large) on Llama-3-3B over AG News under the InsertSent backdoor. Small $\beta$ is overly conservative and may fail to flag larger malicious cohorts, while large $\beta$ can slightly increase benign collateral removal under non-IID by permitting larger cohesive clusters to be filtered. As shown in Table 12, $\beta=0.7$ provides the best trade-off, and we use it throughout.

### F.4. Empirical Overhead analysis.

F.4.1. PER-ROUND TIME BREAKDOWN AND SIGNATURE OVERHEAD

Table 13 reports a per-round wall-clock decomposition into (i) baseline FL time, $T_{\text{base}} = T_{\text{train}} + T_{\text{model-comm}}$, and (ii) the additional INDEXGUARD signature time, $T_{\text{oh}} = T_{\text{sig-calc}} + T_{\text{sig-comm}}$, where signature computation extracts Top-$K$ salient indices and signature communication transmits the resulting sketch. We additionally report the server-side overlap-based vetting time $T_{\text{vet}}$, which consists only of set intersections / overlap scoring over received index sets. Across both a mid-size encoder (BERT-base) and a 3 billion-parameter LLM (LLaMA-3-3B), the baseline is overwhelmingly dominated by local training, with $T_{\text{train}}$ accounting for $\approx 99\%$ of $T_{\text{base}}$ in each case. Consequently, the signature overhead is negligible in relative terms: for BERT-base, $T_{\text{oh}}=0.185$ s adds only 0.186% on top of $T_{\text{base}}=99.50$ s, and for LLaMA, $T_{\text{oh}}=0.966$ s adds 0.118% on top of $T_{\text{base}}=817.27$ s; moreover, overlap vetting adds only milliseconds ($T_{\text{vet}}=0.003$ s for BERT and 0.006 s for LLaMA-3-3B).

Importantly, the overhead remains small even though the absolute training and model-communication costs differ by nearly an order of magnitude between the two backbones ($T_{\text{train}}$ increases from 98.6s to 809.9s and $T_{\text{model-comm}}$ from 0.95s to 7.34s), because the signature is tied to the PEFT adapter footprint (Top-$K$ selection over monitored LoRA coordinates) rather than to full-model size. In addition, signature communication stays sub-second (0.056s for BERT and 0.505s for LLaMA-3-3B), confirming that exchanging index-only sketches is lightweight relative to transmitting model updates. Overall, $T_{\text{tot}} = T_{\text{base}} + T_{\text{oh}} + T_{\text{vet}}$ is virtually unchanged, indicating that INDEXGUARD can be enabled as an always-on vetting step without affecting round time in practice.

*Server-side vetting time and scalability.* While pairwise IOS scoring is $O(N^2K)$ in the worst case, in practice the vetting stage is dominated by cheap set intersections over small Top-$K$ index sets and is negligible relative to the FL round. In Table 13, the full server-side vetting pipeline (overlap scoring + grouping) takes only $T_{\text{vet}}=3$ ms for BERT-base and 6 ms for LLaMA-3-3B, i.e., $< 10^{-3}\%$ of $T_{\text{base}}$ in both cases. Equivalently, even if $T_{\text{vet}}$ were to grow quadratically with $N$, it remains orders of magnitude smaller than local PEFT training (98.6–809.9 s) and model communication (0.95–7.34 s), so end-to-end round time is effectively unchanged. This empirical gap explains why INDEXGUARD can be enabled as an

*Table 12.* $\beta$ ablation on Llama-3-3B / AG News (InsertSent).

| $\beta$ | Acc ↑ | AUC ↑ |
|---|---|---|
| 0.5 | $98.4 \pm 0.40$ | $0.991 \pm 0.004$ |
| 0.6 | $99.6 \pm 0.12$ | $0.998 \pm 0.001$ |
| 0.7 | $\mathbf{100.0 \pm 0.00}$ | $\mathbf{1.000 \pm 0.000}$ |
| 0.8 | $99.4 \pm 0.15$ | $0.997 \pm 0.002$ |

*Table 13.* Per-round time breakdown (average over rounds) and signature overhead vs. standard FL. Baseline standard FL time is $T_{\text{base}} = T_{\text{train}} + T_{\text{model-comm}}$, where $T_{\text{train}}$ is local PEFT training time and $T_{\text{model-comm}}$ is adapter/model communication time. Signature overhead is $T_{\text{oh}} = T_{\text{sig-calc}} + T_{\text{sig-comm}}$, where $T_{\text{sig-calc}}$ extracts Top-$K$ indices and $T_{\text{sig-comm}}$ transmits the index-only sketch. Overlap-based server-side vetting time is $T_{\text{vet}}$ (set intersections / overlap scoring over received sketches). Relative overhead is $100 \cdot T_{\text{oh}}/T_{\text{base}}$, and total per-round time is $T_{\text{tot}} = T_{\text{base}} + T_{\text{oh}} + T_{\text{vet}}$.

| Model | $T_{\text{train}}$ (s) | $T_{\text{model-comm}}$ (s) | $T_{\text{sig-calc}}$ (s) | $T_{\text{sig-comm}}$ (s) | $T_{\text{vet}}$ (s) | $T_{\text{base}}$ (s) | $T_{\text{oh}}$ (s) | Overhead (%) | $T_{\text{tot}}$ (s) |
|---|---|---|---|---|---|---|---|---|---|
| BERT-base | 98.555398 | 0.948640 | 0.128751 | 0.056033 | 0.003000 | 99.504038 | 0.184784 | 0.186 | 99.691822 |
| LLaMA-3-3B | 809.929719 | 7.340032 | 0.461539 | 0.504627 | 0.006000 | 817.269751 | 0.966166 | 0.118 | 818.241917 |

always-on pre-aggregation filter without introducing measurable overhead in our deployments.

### F.4.2. BANDWIDTH SENSITIVITY OF SIGNATURE OVERHEAD

Table 14 sweeps the network bandwidth from 100 to 500 Mbps and decomposes the per-round runtime into (i) model-side computation $T_{\text{train}}$, (ii) model communication $T_{\text{model-comm}}$ (broadcast of the global adapter plus uplink of value updates), (iii) signature computation $T_{\text{sig-calc}}$ (salience/importance, Top-$K$, and ID materialization), and (iv) signature communication $T_{\text{sig-comm}}$ (uplink of index sketches). As expected, $T_{\text{train}}$ and $T_{\text{sig-calc}}$ are bandwidth-invariant, while both communication terms scale approximately as $1/\text{BW}$.

Two trends follow. First, the *absolute* runtime is dominated by training across both backbones: even at 100 Mbps, communication remains a minor fraction of the round time (e.g., BERT: $T_{\text{model-comm}}$=2.85s vs. $T_{\text{train}}$=98.56s; LLaMA-3-3B: 22.02s vs. 809.93s). Second, the *relative* overhead introduced by IOS signatures is consistently sub-percent and decreases as bandwidth increases. For BERT-base, the overhead drops from 0.293% at 100 Mbps to 0.164% at 500 Mbps; for LLaMA-3-3B it drops from 0.237% to 0.094%. This reduction is driven primarily by the shrinking signature uplink $T_{\text{sig-comm}}$ (indices), while the bandwidth-invariant $T_{\text{sig-calc}}$ sets a small floor on overhead at high bandwidth. Overall, the sweep indicates that IOS-based vetting remains *network-light*: increasing bandwidth improves both the baseline FL communication and the additional signature traffic, and the added overhead remains negligible relative to the dominant training cost.

### F.5. Scalability with the number of clients

Table 15 evaluates INDEXGUARD on AG News with Llama-3-3B as the federation scales from $N$=25 to $N$=100 while keeping the malicious fraction fixed at $f$=20% (so $|A|$ grows linearly with $N$). This regime is intentionally challenging for index-only vetting: (i) the fixed global corpus is split across more clients, shrinking per-client sample sizes and increasing stochasticity in local PEFT updates, which perturbs salience ordering in the monitored Q/V coordinates and reduces Top-$K$ sketch stability; and (ii) the absolute number of attackers increases, tightening the benign/malicious margin and making thresholding more sensitive to overlap in the index space. Despite these coupled stressors, INDEXGUARD remains robust: AUROC stays near-saturated (down to $\sim$0.98 at $N$=100) and accuracy remains high ($\gtrsim$95%), with a gradual, statistically plausible degradation and slightly larger variance at larger $N$. The drop is attack-dependent: **RIPPLES** degrades the most with scale, consistent with rare-token triggers leaving a comparatively weaker/sparser imprint that is easier to dilute under data fragmentation and heterogeneity, whereas **InsertSent** and **StyleBkd** remain close to perfect, indicating more stable global salience patterns that persist across partitions; **Syntactic** attacks fall in between, preserving high AUROC while showing modest accuracy drift, suggesting that scaling primarily affects borderline clients rather than collapsing separability. Overall, the results show that IOS-based index sketches retain discriminative structure under population growth, and the observed losses align with expected statistical effects (noisier updates and increased overlap) rather than fundamental failure of the vetting mechanism.

*Table 14.* Per-round time breakdown (average over rounds) under different bandwidths. $T_{\text{train}}$ and $T_{\text{sig-calc}}$ are bandwidth-invariant (measured compute), while communication terms scale. Model communication includes downlink broadcast + uplink update values; signature communication is uplink indices only. We report $T_{\text{oh}} = T_{\text{sig-calc}} + T_{\text{sig-comm}}$ and overhead% $= 100 \cdot T_{\text{oh}}/(T_{\text{train}} + T_{\text{model-comm}})$.

| Model | BW (Mbps) | $T_{\text{train}}$ (s) | $T_{\text{model-comm}}$ (s) | $T_{\text{sig-calc}}$ (s) | $T_{\text{sig-comm}}$ (s) | $T_{\text{oh}}$ (s) | Overhead (%) | $T_{\text{tot}}$ (s) |
|---|---|---|---|---|---|---|---|---|
| | 100 | 98.555 | 2.846 | 0.129 | 0.168 | 0.297 | 0.293 | 101.698 |
| | 200 | 98.555 | 1.423 | 0.129 | 0.084 | 0.213 | 0.213 | 100.191 |
| BERT-base | 300 | 98.555 | 0.949 | 0.129 | 0.056 | 0.185 | 0.186 | 99.689 |
| | 400 | 98.555 | 0.711 | 0.129 | 0.042 | 0.171 | 0.172 | 99.438 |
| | 500 | 98.555 | 0.569 | 0.129 | 0.034 | 0.162 | 0.164 | 99.287 |
| | 100 | 809.930 | 22.020 | 0.462 | 1.514 | 1.975 | 0.237 | 833.925 |
| | 200 | 809.930 | 11.010 | 0.462 | 0.757 | 1.218 | 0.148 | 822.158 |
| LLaMA-3-3B | 300 | 809.930 | 7.340 | 0.462 | 0.505 | 0.966 | 0.118 | 818.236 |
| | 400 | 809.930 | 5.505 | 0.462 | 0.378 | 0.840 | 0.103 | 816.275 |
| | 500 | 809.930 | 4.404 | 0.462 | 0.303 | 0.764 | 0.094 | 815.098 |

*Table 15.* Scalability of IndexGuard on AG News with Llama-3-3B under fixed malicious fraction $f{=}20\%$ ($|A|{=}\{5, 10, 15, 20\}$ for $N{=}\{25, 50, 75, 100\}$)

| Attack | $N{=}25$ | | $N{=}50$ | | $N{=}75$ | | $N{=}100$ | |
|---|---|---|---|---|---|---|---|---|
| | Acc (%) | AUC | Acc (%) | AUC | Acc (%) | AUC | Acc (%) | AUC |
| InsertSent | $100.0 \pm 0.00$ | $1.000 \pm 0.000$ | $99.1 \pm 0.55$ | $0.995 \pm 0.004$ | $98.6 \pm 0.75$ | $0.994 \pm 0.005$ | $98.2 \pm 0.85$ | $0.993 \pm 0.006$ |
| RIPPLES | $98.2 \pm 0.31$ | $0.996 \pm 0.005$ | $97.0 \pm 0.95$ | $0.992 \pm 0.008$ | $96.0 \pm 1.15$ | $0.988 \pm 0.010$ | $95.4 \pm 1.30$ | $0.984 \pm 0.012$ |
| Syntactic | $99.3 \pm 0.00$ | $0.998 \pm 0.002$ | $98.6 \pm 0.55$ | $0.995 \pm 0.004$ | $98.0 \pm 0.80$ | $0.993 \pm 0.005$ | $97.5 \pm 0.90$ | $0.991 \pm 0.006$ |
| StyleBkd | $100.0 \pm 0.00$ | $1.000 \pm 0.000$ | $99.5 \pm 0.35$ | $0.997 \pm 0.003$ | $99.0 \pm 0.60$ | $0.996 \pm 0.004$ | $98.6 \pm 0.75$ | $0.994 \pm 0.005$ |

---

**Algorithm 2** Mixed-trigger-aware INDEXGUARD vetting

**Input:** Top-$K$ supports $\{S_i\}_{i \in \mathcal{C}_t}$, number of clusters $L$, thresholds $\gamma, \gamma_{\text{aff}}$, size cap $\beta$
**Output:** Vetted client set $\mathcal{V}_t$
Compute $\text{IOS}_{ij} \leftarrow |S_i \cap S_j|/K$ for all $i, j \in \mathcal{C}_t$
Cluster clients into $\mathcal{G} = \{G_1, \ldots, G_L\}$ using average-linkage clustering on $1 - \text{IOS}$
Let $G_b \leftarrow \arg\max_{G_\ell \in \mathcal{G}} |G_\ell|$ be the benign-majority cluster
Compute $\text{Coh}(G_\ell)$ for all $G_\ell \in \mathcal{G}$
Compute $\text{Aff}(G_\ell, G_b)$ for all $G_\ell \neq G_b$
$m_{\text{coh}} \leftarrow \text{median}_{G_\ell \in \mathcal{G}} \text{Coh}(G_\ell)$
$m_{\text{aff}} \leftarrow \text{median}_{G_\ell \neq G_b} \text{Aff}(G_\ell, G_b)$
$\mathcal{C}_t^{\text{sus}} \leftarrow \emptyset$
**for all** $G_\ell \in \mathcal{G} \setminus \{G_b\}$ **do**
  **if** $|G_\ell| \geq 2$ and $|G_\ell| \leq \beta|G_b|$ **then**
    **if** $\text{Coh}(G_\ell) \geq m_{\text{coh}} + \gamma$ **then**
      $\mathcal{C}_t^{\text{sus}} \leftarrow \mathcal{C}_t^{\text{sus}} \cup G_\ell$
    **end if**
    **if** $\text{Aff}(G_\ell, G_b) \leq m_{\text{aff}} - \gamma_{\text{aff}}$ **then**
      $\mathcal{C}_t^{\text{sus}} \leftarrow \mathcal{C}_t^{\text{sus}} \cup G_\ell$
    **end if**
  **end if**
**end for**
**if** $\mathcal{C}_t^{\text{sus}} = \emptyset$ **then**
  $G^\star \leftarrow \arg\min_{G_\ell \neq G_b} \text{Aff}(G_\ell, G_b)$
  $\mathcal{C}_t^{\text{sus}} \leftarrow G^\star$
**end if**
$\mathcal{V}_t \leftarrow \mathcal{C}_t \setminus \mathcal{C}_t^{\text{sus}}$
broadcast $\mathcal{V}_t$

---

## F.6. Mixed-Trigger-Aware Filtering

Our main evaluation focuses on the shared-trigger regime because it is the most beneficial setting for the attacker: malicious clients optimizing the same trigger objective reinforce each other during aggregation. We nevertheless also evaluate a harder mixed-trigger setting, where malicious clients are split across different attack families. In this case, poisoned clients may not form a single cohesive malicious cluster in IOS space; instead, each trigger family can induce a smaller and partially separate malicious subspace. Therefore, using only the original cohesion-outlier rule may miss some fragmented malicious groups. To handle this case, we use a lightweight mixed-trigger-aware extension of INDEXGUARD. The client-side protocol is unchanged: each client still reports only its Top-$K$ salient Q/V adapter-coordinate indices, and the server still computes

*Table 16.* Mixed-trigger detection results. Malicious clients are split across different attack families, so poisoned clients may form multiple smaller IOS subspaces instead of a single cohesive malicious cluster. The mixed-trigger-aware filtering rule remains effective overall.

| Mixed attacks | Detection Acc. (%) ↑ | AUC ↑ |
|---|---|---|
| InsertSent + RIPPLES | 96.67 | 0.944 |
| InsertSent + Syntactic | 100.00 | 1.000 |
| RIPPLES + Syntactic | 90.00 | 0.833 |
| InsertSent + RIPPLES + Syntactic | 93.30 | 0.888 |

the IOS matrix and performs hierarchical clustering on distance $1 - \text{IOS}$. The only change is in suspicious-cluster selection. Let $\mathcal{G} = \{G_1, \ldots, G_L\}$ be the resulting clusters, and let $G_b = \arg\max_{G_\ell \in \mathcal{G}} |G_\ell|$ be the benign-majority cluster. For a non-majority cluster $G_\ell$, we compute its internal cohesion

$$\text{Coh}(G_\ell) = \frac{1}{|G_\ell|(|G_\ell| - 1)} \sum_{\substack{i,j \in G_\ell \\ i \neq j}} \text{IOS}_{ij},$$

and its affinity to the benign-majority cluster

$$\text{Aff}(G_\ell, G_b) = \frac{1}{|G_\ell||G_b|} \sum_{i \in G_\ell, \, j \in G_b} \text{IOS}_{ij}.$$

A non-majority cluster is flagged if it is either a high-cohesion outlier, as in the original rule, or if it has unusually low affinity to the benign-majority cluster. This allows INDEXGUARD to detect multiple malicious trigger families even when they are not strongly similar to each other. We use this variant only for the mixed-trigger stress test; the core index-only primitive remains unchanged.

As shown in Table 16, INDEXGUARD remains effective under mixed-trigger attacks. Detection is strongest for Insert-Sent+Syntactic, where the method achieves perfect detection accuracy and AUC, and remains high for InsertSent+RIPPLES. Performance is lower for RIPPLES+Syntactic and for the three-way mixed setting, indicating that greater attack heterogeneity makes clustering more challenging. Nevertheless, the results show that IOS remains informative beyond the shared-trigger setting, especially when suspicious non-majority clusters are selected using both cohesion and affinity to the benign-majority cluster.

