# OpenReview forum: "INDEXGUARD: Index-only Backdoor Vetting for Secure Federated PEFT of Large Language Models"
_ICML.cc/2026/Conference — ICML 2026 regular_

### Official Review · Reviewer_7Uet · 2026-03-05

**Soundness:** 3
**Presentation:** 3
**Significance:** 3
**Originality:** 3
**Overall Recommendation:** 4
**Confidence:** 3

**Summary:**

This paper addresses the critical integrity risk of backdoor attacks in Federated Parameter-Efficient Fine-Tuning (PEFT) for Large Language Models (LLMs). The authors propose INDEXGUARD, an unsupervised, pre-aggregation vetting mechanism designed to identify malicious clients without accessing their real-valued updates, thereby maintaining compatibility with secure aggregation (SecAgg). The core insight is that shared attack objectives induce similar "backdoor imprints" in the salient parameter coordinates of adapters. Clients send only the Top-K indices of their most salient updates; the server then uses Index-Overlap Similarity (IOS) and clustering to filter out cohesive outlier groups (potential attackers) before the final aggregation.

**Compliance With Llm Reviewing Policy:**

Affirmed.

**Final Justification:**

The authors solve most of the problems, so I decide to raise my score. However, I still want to know how zero knowledge proof is specifically implemented in this method.

**Key Questions For Authors:**

When all clients are clean, the expected false positives and their impact on convergence and final server-model performance are unclear; Non-IID data may further exacerbate false positives.


The paper cites both the arXiv and published versions for federated GPT (lines 631–643)


Reference: Index-overlap similarity: A value-free proxy for model relatedness. I couldn't find a publicly published version; is this compliant?

**Limitations:**

Yes

**Strengths And Weaknesses:**

Strengths:

 INDEXGUARD is uniquely "value-free," operating only on indices, which solves a major conflict between backdoor detection and secure aggregation in FL.

 The paper provides a solid mathematical foundation for IOS, analyzing its alignment, stability under noise, and robustness to scaling.

 The method is tested against a wide range of state-of-the-art PEFT variants (LoRA, QLoRA, DoRA, etc.) and diverse textual attacks (RIPPLES, StyleBkd, Syntactic), showing near-perfect AUC in most scenarios.

weaknesses:


When malicious clients embed multiple backdoors (e.g., multi-trigger or multi-target), the method may degrade significantly or even fail.


It is not well established whether a single malicious client can still achieve backdoor injection, whether the method fails in that case, and what the upper bound of tolerable malicious-client ratio is.


Benign clients may cluster naturally by domain/label skew, with cluster sizes comparable to the number of malicious clients, making similarity/cluster-based detection potentially ambiguous.


K is selected based on a warm-up stage, but parameter “significance” can drift during training; detection accuracy/threshold stability may change over many rounds and needs stronger justification.


If attackers know the server-side check, they may manipulate coordinates/significance distributions (e.g., optimizing overlap with the benign reference distribution) and trade off ASR to evade detection, suggesting fragility under informed/adaptive attacks.

---

> ### Author Rebuttal · Authors · 2026-03-31
>
> Dear Reviewer 7Uet,
>
> We thank you for your careful reading and constructive comments. Where some comments overlapped with other reviewers, we use the same or closely aligned response for consistency. We respond to each point below.
>
> **W1-Multi-trigger/Multi-target Attacker.** Our shared-trigger regime assumption is **standard (PEFTGuard, DBA ICLR21) and realistic** because a single adversary controlling multiple malicious clients is more plausible since the adversary benefits from coordinating on the same trigger/objective so their poisoned updates reinforce each other through aggregation to increase its success ratio rather than dilute each other.
> This is also consistent with our observations. Following your suggestion, for instance, on BERT-base/SST-2 with N=10,f=3, when all malicious clients use **InsertSent**, the ASR over the first 5 rounds is **[0.16, 0.28, 0.45, 0.57, 0.82]**. When the 3 malicious clients are split across **InsertSent, RIPPLES, and Syntactic**, the ASR is **[0.04, 0.11, 0.13, 0.14, 0.17]** (we observed the similar trend in other settings). We‘ll clarify this in the revision and add multi-trigger ASR plots to better justify the shared-trigger setting.
>
> **W2-Single malicious client.** Even with only one malicious client, clustering is performed in the IOS distance space, so a sufficiently dissimilar attacker may be separated from the benign population as a small isolated cluster. However, this is a weaker regime than the multi-attacker case, since the strongest signal comes when multiple attackers also form a cohesive outlier group.
> For the tolerable malicious ratio, the relevant upper bound is the usual benign-majority regime, i.e. < 50% malicious clients, because our work is an unsupervised outlier-filtering method and cluster semantics are only identifiable relative to the dominant population. The benign-majority is meaningful since **without trusted labels or a clean reference, unsupervised filtering is only identifiable relative to the dominant population** as shown by many robust FL defenses make the same assumption, including Krum, Trimmed Mean/ Coordinate-wise Median, Bulyan, FLAME, and FoolsGold.
>
> **W3-Benign Cluster Ambiguity.** We agree that strong benign heterogeneity can make clustering more ambiguous, and this is exactly why we evaluate under non-IID Dirichlet partitions and analyze the benign-overlap term ρB​. It is possible for benign clients to form multiple clusters, sometimes with sizes comparable to malicious groups. However, these benign modes can still remain **separable from malicious clusters in IOS space**, so the ambiguity is not purely size-based. Thus, our clustering rule uses relative overlap structure and includes a conservative size cap to avoid over-filtering secondary benign modes. In our results, stronger heterogeneity tightens the benign/poisoned margin, but separation remains strong overall; e.g., in Table 6 on AGNews, when Dirichlet α drops from 0.5 to 0.1, RIPPLES changes from 99.1/0.998 to 97.2/0.989 (Acc/AUC), and Syntactic from 99.6/0.999 to 98.7/0.993.
>
> For LLaMA-7B/AGNews under InsertSent, the detector is essentially perfect at Dir(0.5) (AUC = 1.0, FPR = 0). Even at Dir(0.1), performance remains very strong (AUC = 0.997, TPR = 0.995, FPR = 0.005). Reporting explicit FPR values more systematically would improve clarity. We’ll add them in the revision.
> W4-K Selection/update. K is selected once in warm-up and then fixed, since our analysis shows Top-K supports are stable under bounded rescaling / mild drift. If stronger drift occurs, we can easily recompute K periodically (e.g., every R rounds), as discussed in Appx. A. We’ll expand the discussion on K selection in the revision.
> Adaptive informed attackers. The entire protocol, including the server-side vetting procedure, may be fully known to all clients. This makes early-round protection especially important, since Fig. 1 shows that the backdoor can propagate quickly. Attack detection must operate from the beginning, before malicious influence accumulates. If an informed attacker responds by lying about, diversifying, or fabricating its reported Top-K supports to look benign, by our stronger protected-reporting variant: ZK can be enforced to verify that the reported sketch is correctly derived from the client’s committed local model, preventing arbitrary fake benign-looking reports.
>
> **KeyQ1-FPR under Heterogeneity:** When all clients are clean, FPR should be near zero, not because clustering is trivial, but because our rule only flags non-majority IOS outlier clusters with abnormal internal consistency; that pattern is not expected without a shared malicious imprint. Non-IID benign drift can tighten the margin, but Table 6 shows the degradation remains modest. We’ll add explicit FPR/convergence numbers in the revision.
>
> **KeyQ2:** We’ll fix the duplicate citation in the revision.
>
> **KeyQ3:** This is a concurrent anonymous submission cited per ICML 2026 policy in the supplement.

---

> > ### Author Rebuttal · Reviewer_7Uet · 2026-04-05
> >
> > I appreciate the authors' clarifications, which have resolved my previous doubts. However, I still want to know how zero knowledge proof is specifically implemented in this method.

---

> > > ### Author Response · Authors · 2026-04-05
> > >
> > > Thank you for acknowledging our response.
> > >
> > > As suggested, our zero-knowledge component is implemented as a Groth16 zk-SNARK [Groth, J., 2016], with the circuit written in Circom and setup, proving, and verification handled using snarkjs. The purpose of the proof is to verify the correctness of the reported Top-k support without revealing the underlying client tensor. Concretely, for client i, the private witness is the hidden tensor u_i​, while the public statement consists of the reported Top-K index set S_i together with a public commitment to u_i​. In the current prototype, the indices are public, since this variant models the setting where the server may learn the reported Top-K coordinates while the tensor values remain hidden.
> > >
> > >
> > > The implementation follows the standard offline/online split of Groth16. In the offline phase, the circuit is compiled into an R1CS constraint system and witness generator, and a circuit-specific proving key and verification key are derived from a reusable Powers of Tau file (ptau). This setup is a one-time cost for a fixed circuit parameterization and is then reused across all clients and all FL rounds; it is not repeated per round. In the online phase, each client i computes its Top-K indices locally, generates a witness for its private tensor, proves that the published S_i is correct for that hidden tensor, and sends the proof to the server. The server then verifies the proof using only the public indices S_i, the public commitment, and the verification key; it never sees the plaintext tensor values.
> > >
> > > In our prototype, the circuit proves that the disclosed indices are exactly the Top-K indices of the hidden tensor under a deterministic ranking rule. The compiled circuit contains 6,318 non-linear constraints and 715 linear constraints, with 32 private inputs, 4 public inputs, and 1 public output. **In local runs, proof generation takes about 4.02 s per client, while verification takes about 0.86 s and the offline setup phase takes about 7.1 s, assuming an existing ptau file. As expected for Groth16, proving is the dominant online cost and is paid by the client, whereas verification is substantially cheaper and therefore practical for the server.**
> > >
> > > Overall, **these results show that verifiable Top-K reporting is feasible at prototype scale and can serve as a concrete secure extension of the core INDEXGUARD pipeline. We will include these results in our revision.**
> > >
> > >
> > > [Groth, J., 2016] 2016, April. On the size of pairing-based non-interactive arguments. In Annual international conference on the theory and applications of cryptographic techniques (pp. 305-326). Berlin, Heidelberg: Springer Berlin Heidelberg.]

---

### Official Review · Reviewer_uR9i · 2026-03-11

**Soundness:** 3
**Presentation:** 4
**Significance:** 2
**Originality:** 3
**Overall Recommendation:** 4
**Confidence:** 2

**Summary:**

This paper studies backdoor defense for federated PEFT under secure aggregation. The background is that there might be a malicious client updating the LoRA module while the centralized model can not fully know the detail of each client's update. And their idea is: instead of looking at each client's full update, the server only looks at the indices of the top-K most important adapter coordinates. By measuring how much these index sets overlap across clients, it can filter out suspicious groups before secure aggregation.

**Compliance With Llm Reviewing Policy:**

Affirmed.

**Key Questions For Authors:**

What happens when attackers use different triggers or multiple trigger families in the same round?
Can you explain whether the threat model you proposed is actually realized?

**Limitations:**

The biggest limitation is that the theory/threat model is a bit weak. If the attacker makes some adaptation, how would your system work?

**Strengths And Weaknesses:**

## Strength and weakness

### Strengths

1. The problem framing is clear. Readers can quickly know what the research problem is and what the challenges are. And the problem is important, as it is hard to achieve integrity and confidentiality at the same time.

2. The core idea is clear. The method is easy to understand. It uses only index information, not full values due to the confidentiality. That makes it a natural fit for privacy-friendly federated settings. I also like that the method is simple enough that people could actually try it in practice.

3. The results are strong on the paper's chosen setup. On the tasks and attacks they test, the method seems to work well. The reported AUC numbers are high, and the defense reduces the attack success rate while keeping clean accuracy fairly stable. That is a good sign.

4. The overhead looks small. The extra computation and communication seem light.

---



### Weaknesses
1. The threat model is narrow. This is my biggest concern. The method seems to rely on malicious clients behaving in a similar way. In other words, they need to leave a similar footprint in the top-K coordinate indices. That may hold for a shared-trigger setting, but it is not clear how well this will hold under stronger attackers.

For example, what happens if attackers:
- use different triggers,
- spread out their important coordinates on purpose,
- pad their top-K sets with benign-looking indices?

Or what if there is only one attacker, so there is no overlap with other attackers?

These are natural adaptive strategies. The current paper does not really answer this.

2. The theory is limited

The theory gives intuition for why overlap should be higher among malicious clients. That is useful. But the assumptions are quite favorable to the method. In effect, the theory assumes that attackers tend to share a trigger-related subset that appears in their top-K coordinates with high probability.

So the theory mostly explains the setting where the method should work. It does not give a strong account of when it will fail or how robust it is if attackers adapt. And also, I don't think this theory/threat model may be realized.

---

> ### Author Rebuttal · Authors · 2026-03-31
>
> Dear Reviewer uR9i,
>
> We sincerely thank you for your careful reading and constructive comments. Where some comments overlapped with other reviewers, we use the same or closely aligned response for consistency.  We respond to each point below.
>
>
> **W1-Attacker Model:**
>
> **Trigger Regime.** Our shared-trigger regime assumption is **standard (PEFTGuard, DBA ICLR21) and realistic** because a single adversary controlling multiple malicious clients is more plausible since the adversary benefits from coordinating on the same trigger/objective so their poisoned updates reinforce each other through aggregation to increase its success ratio rather than dilute each other.
> This is also consistent with our observations. Following your suggestion, for instance, on BERT-base/SST-2 with N=10,f=3, when all malicious clients use **InsertSent**, the ASR over the first 5 rounds is **[0.16, 0.28, 0.45, 0.57, 0.82]**. When the 3 malicious clients are split across **InsertSent, RIPPLES, and Syntactic**, the ASR is **[0.04, 0.11, 0.13, 0.14, 0.17]** (we observed the similar trend in other settings). We will clarify this in the revision and add multi-trigger ASR plots to better justify the shared-trigger setting.
>
> **Adaptive Attacks.** We agree that a stronger adaptive attacker could try to evade INDEXGUARD by **changing its training behavior so that true salient coordinates are more spread out/benign-looking**. This is a limitation since our theory characterizes the regime where index-overlap vetting is identifiable, not robust against all adaptive attackers. However, such adaptation isn’t cost free. If the attacker weakens the IOS signal, it also weakens/distorts the shared salient-coordinate structure making the backdoor effectively propagate quickly from early rounds (see Fig. 1).
> Separately, if the attackers try to evade detection by misreporting Top-K supports rather than changing the actual optimization, **this is the integrity issue addressed by our stronger secure variant in Step 4.** Clients can submit protected reports together with ZK proof that the server can verify that the reported Top-K sketch is actually derived from the client’s committed local model, rather than being an arbitrary fake report. ZK becomes a natural additional safeguard that can be applied each round. However, we also implemented ZK-proof as a prototype where proof generation takes ~4.02s and verification takes ~0.86s suggesting that verifiable index reporting is practical.
>
> **Single Attacker.** A singleton attacker may still be detectable because we use its distance/overlap relative to benign set, not only non-majority cluster cohesion; however, the signal is strongest when multiple attackers form a dense malicious cluster. This is also consistent with Table 10, where detection improves slightly as the number of malicious clients increases.
>
> **W2-Theory:** Our theory focuses on the regime that is both practically meaningful and attacker-favorable. When attackers use different triggers/behaviors, the malicious updates become less aligned, so propagation through aggregation is weaker than in the coordinated shared-trigger case. In that sense, this is a harder case for detection, but also a **less effective attack regime.**
> We agree that the theory is intentionally scoped. Its role is to formalize the regime where index-overlap vetting is identifiable, not to prove robustness against all adaptive attacks. In particular, the theory assumes a shared malicious imprint in salient coordinates and therefore explains when INDEXGUARD should succeed, rather than characterizing every possible failure mode.
>
> **Adaptive Attacks:** That said, more adaptive attackers are not cost-free in our setting. If an attacker deliberately changes its optimization to evade the IOS signal, it must reduce or distort the very salient-coordinate structure that helps the backdoor propagate effectively through aggregation, so adaptation naturally introduces an **evasion–attack-strength tradeoff**. For dishonest adaptation at the reporting level, our stronger protected-reporting variant with zero-knowledge can be enforced each round to prevent clients from lying, fabricating, or artificially diversifying their reported Top-K supports. We will clarify this scope more explicitly in the revision.
>
> **KeyQ1:** If attackers use different triggers in the same round, they may split into smaller IOS subgroups instead of one cohesive malicious cluster, weakening detection. Since INDEXGUARD **uses both relative distance/overlap and non-majority cluster cohesion, realizing our threat model**, it may still separate small malicious subgroups, but the signal is weaker because each subgroup has fewer attackers. We focus on the shared-trigger setting because a coordinated adversary controlling multiple malicious clients is both realistic and attacker-favorable

---

> > ### Author Rebuttal · Reviewer_uR9i · 2026-03-31
> >
> > I still have concerns about the scope of this paper, which might be restricted to a certain type of attack. But this paper is doing well under that assumption.

---

> > > ### Author Response · Authors · 2026-04-05
> > >
> > > We thank the reviewer uR9i for the careful follow-up. To address the remaining concerns as directly as possible, we now distinguish two stronger settings supported by further experimental results that will be added in the revised version: 1) mixed-trigger regime beyond shared-trigger; and 2) adaptive reported-support manipulation.
> > >
> > > **In the mixed-trigger setting**, malicious clients need not form one cohesive malicious cluster and may instead split into smaller IOS subspaces as we discussed in W1-Attacker Model. Different attackers employ different attack families simultaneously (e.g., InsertSent, RIPPLES, and Syntactic). To study this harder regime, we extended ours with a lightweight mixed-attack-aware filtering rule: clients still send only Top-K indices and the server still performs IOS-based pre-aggregation clustering, but suspicious non-majority clusters are flagged not only by cohesion but also by weak affinity to the benign-majority cluster.
> > >
> > > Mixed attacks| Acc | AUC
> > > :--|--:|--:|
> > > InsertSent + RIPPLES | 96.67% | 0.944
> > > InsertSent + Syntactic | 100% | 1.0
> > > RIPPLES + Syntactic | 90.0% | 0.833
> > > InsertSent+ RIPPLES + Syn | 93.3% | 0.888
> > >
> > > **The results show that IndexGuard remains effective under mixed-trigger regime.** Detection is strongest for InsertSent + Syntactic, which achieves perfect acc and AUC, while InsertSent + RIPPLES also performs very well. Performance slightly drops for RIPPLES + Syntactic and for the three-way mixed setting, indicating that greater attack heterogeneity makes clustering more challenging. Nevertheless, **results remain strong overall, suggesting that the IOS is still informative even beyond the shared-trigger setting.**
> > >
> > >
> > > **Adaptive attack.**  If a malicious client submits an arbitrary benign-looking Top-K support Si that is not derived from its local update, this is exactly the integrity gap addressed by our protected-reporting variant: with ZK proof, the server can verify that the reported Top-K sketch is correctly derived from the client’s committed local model. To quantify the difficulty of such evasion before ZK is enforced, we evaluated a stronger dishonest-reporting attacker that trains a poisoned local model but transmits a Top-K support obtained by mixing clean and poisoned supports so as to look more benign.
> > >
> > > b ratio | INDEXGUARD  Acc. (%) | INDEXGUARD AUC | ASR@10 (%, no mitigation)
> > > :--|--:|--:|--:
> > > 0.1 | 71.2 | 0.55 | 55.1
> > > 0.2 | 75.8 | 0.66 | 58.3
> > > 0.3 | 77.7 | 0.69 | 62.6
> > > 0.4 | 96.7 | 0.94 | 66.5
> > > 0.5 | 100 | 1.0 | 82.4
> > >
> > > The results show a clear attack-strength tradeoff: for b=0.1,0.2,0.3 (b is the fraction of non-overlapping Top-K indices drawn from the poisoned support; smaller b makes the transmitted support more benign-looking), IndexGuard achieves 71.2/0.55, 75.8/0.66, and 77.7/0.69 (Acc/AUC), while ASR@10 without mitigation is only 55.1, 58.3, 62.6, respectively. For b=0.4,0.5, ASR rises to 66.5 and 82.4, but detection becomes much easier, reaching 96.7/0.94 and 100/1.0. Thus, **before protected reporting is enforced, the attacker can either remain less detectable with limited ASR benefit, or gain stronger ASR at the cost of becoming highly detectable; once ZK is enforced, this attack surface is removed entirely.**

---

### Official Review · Reviewer_SXQQ · 2026-03-11

**Soundness:** 2
**Presentation:** 2
**Significance:** 2
**Originality:** 2
**Overall Recommendation:** 3
**Confidence:** 3

**Summary:**

This paper studies backdoor defense for federated PEFT of large language models under privacy-preserving secure aggregation, where the server cannot inspect full client updates. To address this problem, the paper proposes INDEXGUARD, an index-only client vetting method that asks each client to send only Top-K salient adapter-coordinate indices, then uses overlap-based similarity and clustering to detect suspicious malicious groups before aggregation. The paper evaluates the method across multiple backbones, PEFT variants, attack settings, and non-IID scenarios and shows that it can reduce attack success rate while largely keeping clean accuracy.

**Compliance With Llm Reviewing Policy:**

Affirmed.

**Final Justification:**

This paper studies a meaningful setting and the index-only design is clean, but I do not think the rebuttal fully resolves the main concern: the method still rests on fairly narrow assumptions, and the paper does not directly show robustness against adaptive attackers who deliberately diversify or pad their Top-K supports to evade clustering while maintaining backdoor strength. The work would be much stronger with direct adaptive-attack experiments under matched settings, broader tests beyond the shared-trigger/benign-majority regime, and stronger head-to-head comparison against the best federated backdoor defenses that are as close as possible to this secure-aggregation setting, so I would maintain my score.

**Key Questions For Authors:**

1. How does INDEXGUARD perform against adaptive attackers that intentionally change their Top-K supports to evade overlap-based clustering?

2. Can the authors provide more experiments beyond the shared-trigger and benign-majority regimes?

3. How practical is the stronger secure version with protected indices and zero-knowledge proofs?

4. Can the authors include more direct empirical comparison with strong federated backdoor defenses under comparable settings?

5. What is the false positive rate under strong benign heterogeneity?

**Limitations:**

Yes

**Strengths And Weaknesses:**

Strengths

1. The setting is meaningful because many existing federated defenses require access to full client updates, but this is not possible under secure aggregation. The paper targets this gap directly.

2. Using only Top-K salient indices is a clean design choice. It matches the privacy constraint better than many prior defenses that need real-valued update inspection.

3. The paper tests multiple models, multiple PEFT methods, several attacks, and non-IID settings. This gives decent evidence that the method is not only working in one narrow case.

4. The paper is generally clear. The workflow is easy to follow, and the motivation of index-only vetting is presented clearly.

Weaknesses

1. The method depends on several strong assumptions. A key assumption is that malicious clients share similar trigger behavior, so their salient indices become more similar. Another important assumption is that the largest cluster is the benign-majority cluster. These assumptions may not hold in more adaptive or difficult settings.

2. The adaptive attacker story is still not strong enough. It is not fully clear how well INDEXGUARD works if attackers intentionally diversify their Top-K supports to avoid clustering while still preserving backdoor strength.

3. The full secure-deployment story feels incomplete. The paper discusses stronger protection like protected reporting and zero-knowledge proofs, but this seems more like an additional future extension than a fully validated part of the current system.

4. Baseline comparison can be stronger. Although the paper explains why many prior methods do not fit this setting, I still want more direct empirical comparison with the strongest possible federated backdoor defenses under matched settings.

5. The paper mentions limitations such as majority-malicious settings and need for shared sketch space, but these points are important enough to be highlighted more directly in the main discussion.

---

> ### Author Rebuttal · Authors · 2026-03-31
>
> Dear Reviewer SXQQ,
> We sincerely thank you for your careful reading and constructive comments. We respond to each point below.
>
> **W1/KeyQ2-Adversarial Assumptions.** Our shared-trigger regime assumption is **standard (PEFTGuard, DBA ICLR21) and realistic** because a single adversary controlling multiple malicious clients is more plausible since the adversary benefits from coordinating on the same trigger/objective so their poisoned updates reinforce each other through aggregation to increase its success ratio rather than dilute each other.
> This is also consistent with our observations. Following your suggestion, for instance, on BERT-base/SST-2 with N=10,f=3, when all malicious clients use **InsertSent**, the ASR over the first 5 rounds is **[0.16, 0.28, 0.45, 0.57, 0.82]**. When the 3 malicious clients are split across **InsertSent, RIPPLES, and Syntactic**, the ASR is **[0.04, 0.11, 0.13, 0.14, 0.17]** (we observed the similar trend in other settings). The shared-trigger propagates much more effectively, while the mixed-trigger case is weaker since the malicious updates are less aligned and are diluted by aggregation. We will clarify this in the revision and add multi-trigger ASR plots to better justify the shared-trigger setting.
> The benign-majority is the usual operating setting in FL for a diverse set of unsupervised/robust defenses (Krum, Trimmed Mean/Coordinate-wise Median, Bulyan, FLAME, and FoolsGold). It is meaningful because **without trusted labels or a clean reference, unsupervised filtering is only identifiable relative to the dominant population**. Therefore, malicious clients are treated explicitly or implicitly as a minority/outlier group relative to benign clients.
>
> **W2/KeyQ1-Adaptive Attack/Diversifying Top-K.** We agree that a stronger adaptive attacker could try to evade INDEXGUARD by **changing its training behavior so that true salient coordinates are more spread out/benign-looking**. This is a limitation since our theory characterizes the regime where index-overlap vetting is identifiable, not robust against all adaptive attackers. However, such adaptation isn’t cost free. If the attacker weakens the IOS signal, it also weakens/distorts the shared salient-coordinate structure making the backdoor effectively propagate quickly from early rounds (see Fig. 1).
> Separately, if the attackers try to evade detection by misreporting Top-K supports rather than changing the actual optimization, this is the integrity issue addressed by our stronger secure variant in Step 4. Clients can submit protected reports together with ZK proof that the server can verify that the reported Top-K sketch is actually derived from the client’s committed local model, rather than being an arbitrary fake report.
>
> **W3-Full Secure Deployment.** The protected-reporting/ZK layer is best viewed as an extension for stricter adversarial deployments. For example, in many practical cross-silo FL settings, participants are contracted or authenticated entities following the protocol, while poisoning may still enter through upstream data collection or annotation pipelines; in this case, clients do not lie about their Top-K reports, so our core pipeline is already the relevant deployment model. By contrast, in a stronger adversarial setting where clients may fabricate their reported supports, ZK becomes a natural additional safeguard that can be applied each round. However, we also implemented ZK-proof as a prototype where proof generation takes ~4.02s and verification takes ~0.86s suggesting that verifiable index reporting is practical.
>
> **W4-Baseline Comparison**. Other FL backdoor studies (e.g., AlignIns, DBA) developed for conventional FL backdoor detection on different settings (e.g., vision-style/image backdooring), rather than federated PEFT of LLMs. Therefore, comparing them to ours not only requires structural changes but also needs to modify their key components. After our submission, ProtegoFed (ArXiv) was proposed for federated instruction tuning. That being said, it still assumes access to richer shared training signals than our index-only secure-aggregation setting. We’ll add a comparison to ProtegoFed in the revision to assess robustness against value-sharing backdoor defenses.
>
> **W5-Highlighting Malicious Majority**. Thank you for the suggestion. We’ll revise the paper to highlight it more directly in the main discussion.
>
> **KeyQ5-FPR under benign heterogeneity**. We agree that explicit FPR numbers would improve clarity. We currently report AUC under non-IID Dirichlet partitions rather than a single thresholded FPR, since AUC captures the full TPR–FPR tradeoff under heterogeneity. Consistent with Table 6, for LLaMA-7B/AGNews under InsertSent, with Dir(0.5) the detector achieves essentially perfect separation (AUC = 1.0, FPR = 0). When heterogeneity increases to Dir(0.1), performance remains very strong, with AUC = 0.997, TPR = 0.995, and FPR = 0.005. We will add explicit FPR numbers under heterogeneity.

---

> > ### Author Rebuttal · Reviewer_SXQQ · 2026-04-04
> >
> > The rebuttal improves the paper, especially by clarifying the deployment story and giving more explicit evidence on false positives under heterogeneity, but it still does not fully resolve the main concerns. The biggest remaining weakness is robustness to adaptive attackers: the response argues why diversification may be costly for the attacker, but it does not provide direct experiments showing INDEXGUARD still works when attackers intentionally change their salient Top-K supports to evade clustering. The assumptions behind the method also remain somewhat narrow, and the baseline comparison is still not as strong or as direct as it should be under matched settings. Overall, I would like to maintain my score.

---

> > > ### Author Response · Authors · 2026-04-05
> > >
> > > We thank the reviewer SXQQ for the careful follow-up. To address the remaining concerns as directly as possible, we now distinguish three stronger settings supported by further experimental results that will be added in the revised version: 1) mixed-trigger regime beyond shared-trigger; 2) adaptive reported-support manipulation; and 3) a stronger FL baseline comparison.
> > >
> > >
> > > **In the mixed-trigger setting**, malicious clients need not form one cohesive malicious cluster and may instead split into smaller IOS subspaces as we discussed in W1/KeyQ2-Adversarial Assumptions. Different attackers employ different attack families simultaneously (e.g., InsertSent, RIPPLES, and Syntactic). To study this harder regime, we extended ours with a lightweight mixed-attack-aware filtering rule: clients still send only Top-K indices and the server still performs IOS-based pre-aggregation clustering, but suspicious non-majority clusters are flagged not only by cohesion but also by weak affinity to the benign-majority cluster.
> > >
> > > Mixed attacks | Acc | AUC
> > > :--|--:|--:
> > > Ins + RIP | 96.67% | 0.944
> > > Ins + Synt | 100% | 1.0
> > > RIP + Syn | 90.0% | 0.833
> > > Ins + RIP + Syn | 93.3% | 0.888
> > >
> > > The results show that IndexGuard remains effective under mixed-trigger regime. Detection is strongest for Ins+Syn, which achieves perfect acc and AUC, while Ins+RIP also performs very well. Performance slightly drops for RIP+Syn and for the three-way mixed setting, indicating that greater attack heterogeneity makes clustering more challenging. Nevertheless, results remain strong **overall, suggesting that the IOS is still informative even beyond the shared-trigger setting.**
> > >
> > >
> > >
> > > **Adaptive attack.**  If a malicious client submits an arbitrary benign-looking Top-K support Si that is not derived from its local update, this is exactly the integrity gap addressed by our protected-reporting variant: with ZK proof, the server can verify that the reported Top-K sketch is correctly derived from the client’s committed local model. As suggested by the reviewer, to quantify the difficulty of such evasion before ZK is enforced, we evaluated a stronger dishonest-reporting attacker that trains a poisoned local model but transmits a Top-K support obtained by mixing clean and poisoned supports so as to look more benign.
> > >
> > > b ratio | Acc. (%) | AUC | ASR@10 (%, no mitigation)
> > > :--|--:|--:|--:
> > > 0.1 | 71.2 | 0.55 | 55.1
> > > 0.2 | 75.8 | 0.66 | 58.3
> > > 0.3 | 77.7 | 0.69 | 62.6
> > > 0.4 | 96.7 | 0.94 | 66.5
> > > 0.5 | 100 | 1.0 | 82.4
> > >
> > > The results show a clear attack-strength tradeoff: for b=0.1,0.2,0.3 (b is the fraction of non-overlapping Top-K indices drawn from the poisoned support; smaller b makes the transmitted support more benign-looking), Ours achieves 71.2/0.55, 75.8/0.66, and 77.7/0.69 (Acc/AUC), while ASR@10 without mitigation is only 55.1, 58.3, 62.6, respectively. For b=0.4,0.5, ASR rises to 66.5 and 82.4, but detection becomes much easier, reaching 96.7/0.94 and 100/1.0. Thus, **before protected reporting is enforced, the attacker can either remain less detectable with limited ASR benefit, or gain stronger ASR at the cost of becoming highly detectable; once ZK is enforced, this attack surface is removed entirely.**
> > >
> > >
> > > **Extended Baseline Comparison.** We also strengthened the baseline analysis. At the time of our submission, to the best of our knowledge, there was no prior FL backdoor detection method specifically for FL on PEFT adaptation. To address your concern (compare against the closest FL LLM backdoor defense), we added an auxiliary comparison to ProtegoFed (arxiv, appeared after our submission). ProtegoFed is not a direct alternative to ours, as the client shares values with the server (different observability regime from our index-only secure-aggregation). This comparison is not fully **apples-to-apples**: ours is a client-level, index-only pre-aggregation vetting method for the malicious-client setting, whereas ProtegoFed is a sample-level filtering that is more natural when poisoned samples are interspersed across otherwise benign clients. We adapted them as an auxiliary overlap comparison using BERT-base-uncased/SST-2 (N=30, f=0.3) and reported their CA and ASR.
> > >
> > > Method | CA↑ | ASR↓ | Early ASR behavior↓|
> > > :--|--:|--:|--:
> > > Ours | 0.59 | 0.085 | 0.018 at R2
> > > ProtegoFed | 0.50 | 0.000 | 1.00 during R1–R3
> > >
> > > The result  shows a clear tradeoff. **Ours achieves substantially better final CA than ProtegoFed, while also suppressing the attack much earlier in training.** In contrast, ProtegoFed method eventually attains the strongest final attack removal (ASR = 0 at Round 10), but only after several rounds with very high vulnerability, and with noticeably lower clean utility overall. This results shows that our method is particularly strong at early and utility-preserving mitigation, even though a sample-level filtering method can reach a lower final ASR in its more natural regime.

---

### Decision · Program_Chairs · 2026-04-30

**Decision:**

Accept (regular)

**Comment:**

This paper proposes INDEXGUARD, a novel technique for detecting backdoor attacks on PEFT in a federated setting by measuring the top-K most important indices from each client, but without assuming access to raw model updates. The reviewers appreciated the novel setting, and the observation that update indices can be indicative of collusion. The reviewers made several suggestions for strengthening the paper, including adding more experiments on adversarial attackers and comparing against more recent, SOTA defenses. The authors have done these things during the rebuttal period, including showing the tradeoff between ASR and INDEXGUARD detection rates as the attackers attempt to mix their malicious updates with benign ones. They also compared their method against ProtegoFed, which came out after the ICML deadline and also has a weaker threat model. Finally, they added experiments on the mixed-trigger setting. Overall, the paper proposes a new angle on the very well-studied problem of detecting backdoor attacks in FL deployments, and demonstrates one viable defense.